# Structural basis of TMPRSS11D specificity and autocleavage activation

Bryan J. Fraser [1,2] ✉, Ryan P. Wilson[1], Sára Ferková[3,4], Olzhas Ilyassov[1], Jackie Lac[1], Aiping Dong[1], Yen-Yen Li [1], Alma Seitova[1], Yanjun Li [1], Zahra Hejazi[1], Tristan M. G. Kenney [2,5], Linda Z. Penn[2,5], Aled Edwards [1,2], Richard Leduc [3,4], Pierre-Luc Boudreault [3,4], Gregg B. Morin [6,7,8] ✉, François Bénard[7,8] ✉ & Cheryl H. Arrowsmith [1,2,5] ✉

Transmembrane Protease, Serine-2 (TMPRSS2) and TMPRSS11D are human proteases that enable SARS-CoV-2 and Influenza A/B virus infections, but their biochemical mechanisms for facilitating viral cell entry remain unclear. We show these proteases spontaneously and efficiently cleave their own zymogen activation motifs, activating their broader protease activity on cellular substrates. We determine TMPRSS11D co-crystal structures with a native and an engineered activation motif, revealing insights into its autocleavage activation and distinct substrate binding cleft features. Leveraging this structural data, we develop nanomolar potency peptidomimetic inhibitors of TMPRSS11D and TMPRSS2. We show that a broad serine protease inhibitor that underwent clinical trials for TMPRSS2-targeted COVID-19 therapy, nafamostat mesylate, was rapidly cleaved by TMPRSS11D and converted to low activity derivatives. In this work, we develop mechanistic insights into human protease viral tropism and highlight both the strengths and limitations of existing human serine protease inhibitors, informing future drug discovery efforts targeting these proteases.

Human respiratory viruses pose significant threats to global public health. The emergence of Severe Acute Respiratory Syndrome Coronavirus-2 (SARS-CoV-2) and the COVID-19 pandemic has highlighted the urgent need for antiviral therapeutics that can be deployed when respiratory virus vaccines are not available. To this end, the viral entry mechanism of SARS-CoV-2 has been intensely studied to identify druggable protein targets for SARS-CoV-2 and other human coronavirus infections[1–5]. A family of cell surface human proteases, the Type II Transmembrane Serine Proteases (TTSPs), have been shown to drive efficient SARS-CoV-2 viral entry and infection and are important drug targets for host-targeted antiviral prophylactics and/or

therapeutics[3,5–8]. Furthermore, the TTSPs have been shown to play critical roles in cancer aggressiveness and metastasis when their proteolytic activity becomes dysregulated, motivating the development of TTSP-targeted anti-cancer agents[9–13].

TTSPs are first produced as inactive precursors (zymogens). Proteolytic cleavage at a specific (Arg/Lys)-(Ile/Val) peptide bond in their activation motif activates their serine protease (SP) domains, enabling them to proteolyze cellular substrates[14–19] (Fig. 1a). Thus, zymogen cleavage activation is the most significant post-translational modification for TTSP enzymatic activity and biological function. Functionally important substrates of the SP domains of TTSPs include

[1]Structural Genomics Consortium Toronto, Toronto, ON, Canada. [2]Department of Medical Biophysics, University of Toronto, Toronto, ON, Canada. [3]Department of Pharmacology and Physiology, Faculty of Medicine and Health Sciences, Institut de Pharmacologie de Sherbrooke, Université de Sherbrooke, Sherbrooke, QC, Canada. [4]Department of Radiology, University of British Columbia, Vancouver, BC, Canada. [5]Princess Margaret Cancer Centre, Toronto, ON, Canada. [6]Canada's Michael Smith Genome Sciences Centre, Vancouver, BC, Canada. [7]British Columbia Cancer Research Institute, Vancouver, BC, Canada. [8]University of British Columbia, Vancouver, BC, Canada. ✉e-mail: Bryanj.fraser@utoronto.ca; gmorin@bcgsc.ca; fbenard@bccrc.ca; Cheryl.arrowsmith@uhn.ca

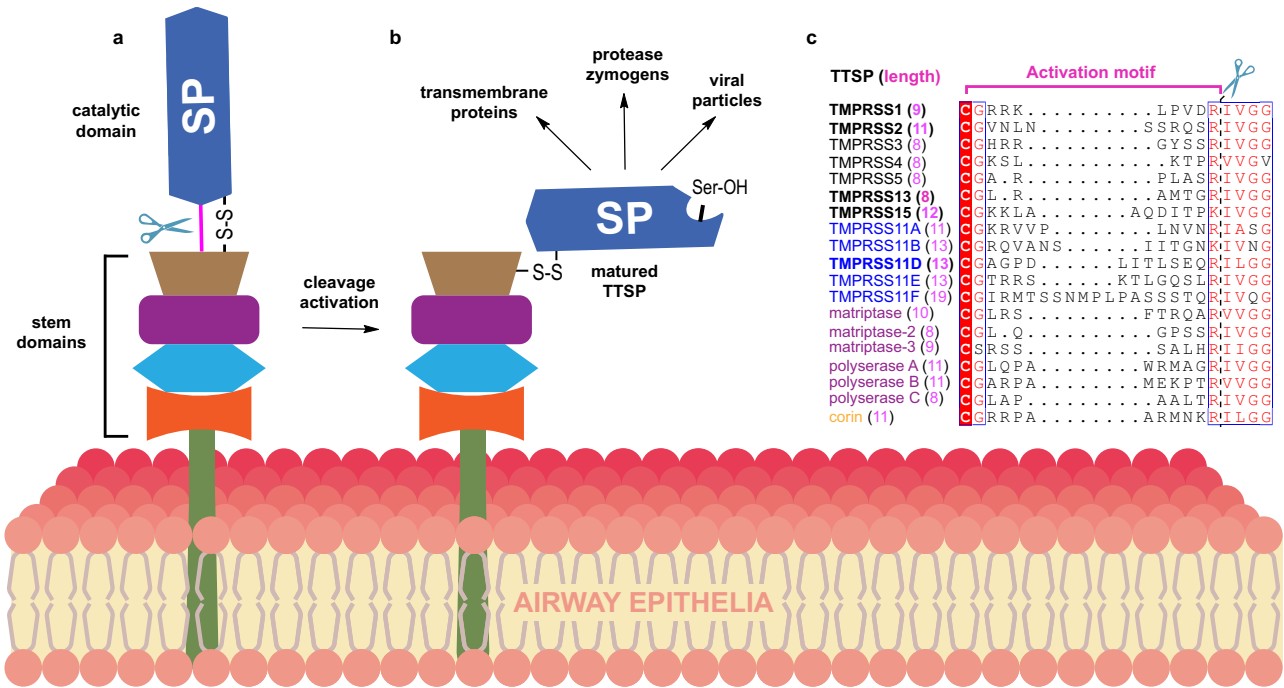

**Fig. 1 | The zymogen activation motif of TTSPs are cleaved by trypsin-like serine proteases and their residue composition is distinct to each TTSP. a** Schematic of an inactive (zymogen) TTSP at the cell surface. The catalytic Serine Protease (SP) domain is connected to the non-catalytic (stem) domains through a disulfide bond (S-S) and the zymogen activation motif peptide bond, shown as a pink line. The zymogen motif peptide bond is cleaved (indicated with scissors) to form (**b**) the matured TTSP that has enzymatic activity and can cleave protein and/or peptide substrates. **c** Multiple sequence alignment of the zymogen activation motif of all human TTSPs. TTSPs are colored by TTSP subfamily; hepsin/TMPRSS-black; HAT/DESC-blue; matriptase-magenta; corin-orange. The length of the zymogen activation motif is indicated in parentheses for each TTSP. Scissors and a dashed black line indicate where TTSPs are cleaved during protease zymogen activation.

membrane proteins, extracellular matrix proteins, hormone precursors, protease zymogens and viral particles (Fig. 1b)[20]. One of the most intensely studied TTSPs, Transmembrane protease, serine-2 (TMPRSS2), cleaves the SARS-CoV-2 Spike protein to enable cell entry by the virus[3,8,21–23]. Other TTSPs have also been shown to play critical roles in SARS-CoV-2 and influenza virus infections in the absence of TMPRSS2[24–26]. TMPRSS11D (Human Airway Trypsin-like protease; HAT; Uniprot O60235), a member of the HAT/DESC subfamily of TTSPs, is highly expressed in the human airways and can enable SARS-CoV-2 and Influenza A infection[24,27–29]. The protein domain organization of TMPRSS11D (and all other HAT/DESC subfamily members) consists of a small cytoplasmic domain at the N-terminus, a single-pass transmembrane domain, a Sea urchin, Enteropeptidase and Agrin (SEA) domain, and a C-terminal SP domain. The only reported protein crystal structure of the HAT/DESC subfamily is the SP domain of TMPRSS11E (DESC1; Uniprot Q9UL52)[30] and no selective inhibitors of any HAT/DESC subfamily members have been described to date. Furthermore, the high sequence similarity shared amongst the HAT/DESC group suggests that they may have similar protein substrate preferences, and it is unclear what interconnected role(s) these proteases have and any potential redundancy they have within the human airways.

In this study, we applied a protein engineering method that enabled the high-yield production of active TMPRSS11D protease and determined X-ray crystal structures of the TMPRSS11D serine protease domain bound to a native, cleaved product molecule. We employed biochemical assays and developed peptidomimetic inhibitors that demonstrate TMPRSS2 and TMPRSS11D recognize and cleave their own zymogen activation motifs to turn on their proteolytic activity, potentially explaining why their protease activities are exploited by respiratory viruses for viral entry. Through a combination of biochemical and biophysical assays, crystal structures, computational modeling, and peptidomimetic inhibitor development, we gained

insights into TMPRSS11D substrate and inhibitor recognition and nafamostat inhibitor binding kinetics. We provide tools to modulate the activities of important human proteases involved in respiratory viral infections, including those caused by emerging respiratory viruses.

## Results

### Autocleavage limits TMPRSS2 and TMPRSS11D protein yield

We previously acquired a highly active source of the TMPRSS2 ectodomain using a directed activation strategy (das) that replaced the TMPRSS2 zymogen activation motif with DDDDK[255] ↓ IVGG (dasTMPRSS2). This strategy enables control of cleavage activation after protein purification from baculovirus-infected Sf9 insect cells[15]. Without the DDDDK sequence replacement or a catalytic serine mutation (S441A), we were unable to overexpress any soluble TMPRSS2 protein. Others have since replicated production and purification of the dasTMPRSS2 protein to determine various TMPRSS2 crystal structures[31,32]. We reapplied this approach to TMPRSS11D by replacing its zymogen activation motif, LSEQR[186] ↓ ILGG with DDDDK[186] ↓ ILGG to create a dasTMPRSS11D construct, which greatly improved protein expression levels relative to the wild-type TMPRSS11D ectodomain protein which showed no detectable protein band (eTMPRSS11D; Fig. 2a). To study the TMPRSS11D ectodomain with its native zymogen activation motif, we introduced a catalytic serine mutation, S368A, which improved protein expression levels (eTMPRSS11D S368A; Fig. 2b). These protein expression data indicated that constitutive TMPRSS11D protease activity poses challenges to its overexpression in recombinant host cells, similar to TMPRSS2[15].

When the dasTMPRSS11D protein was overexpressed, purified, and concentrated to ~5 mg/mL, it underwent rapid autocleavage activation (Fig. 2c). The activated dasTMPRSS11D protein migrated on SDS-PAGE gels at a molecular weight of approximately 27 kDa and produced a single elution peak when purified further by size-exclusion

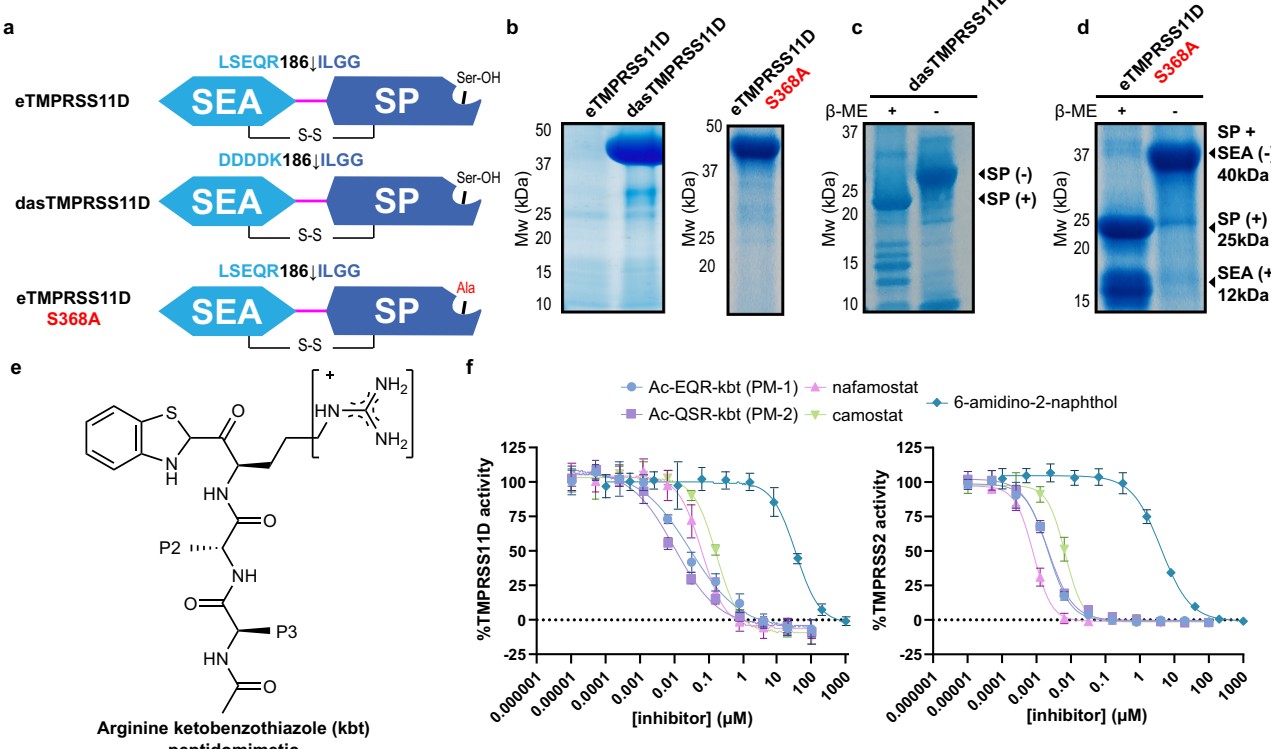

**Fig. 2 | Active soluble TMPRSS11D is accessible by replacing its zymogen activation motif with a DDDDK sequence and its activity is blocked by peptidomimetic and small molecule inhibitors. a** Schematics of soluble human TMPRSS11D protein constructs that span the TMPRSS11D ectodomain. The two protein domains include the Sea urchin, Enteropeptidase and Agrin (SEA) and Serine Protease (SP) domains which are connected by a disulfide bond (S-S) and the R186-I187 peptide bond. Mutations targeting the R186-I187 cleavage site for each protein construct are indicated. **b** TMPRSS11D protein test expression studies from baculovirus-infected Sf9 insect cells. The indicated TMPRSS11D protein was purified from media through IMAC purification and protein content evaluated by SDS-PAGE. Samples were thermally denatured and reduced (4x Laemmli buffer containing 5 mM β-mercaptoethanol, 95 °C, 5 min) prior to gel separation. **c** Purified, active dasTMPRSS11D protein. SDS-PAGE samples were thermally denatured and reduced (+) or were not heated and not reduced (−) in advance of gel separation.

**d** Purified, activated eTMPRSS11D S368A protein. All protein gel images (**b**–**d**) are representative of $n \geq 3$ independent biological experiments. **e** Chemical structure of an arginine ketobenzothiazole (kbt) peptidomimetic inhibitor. A ligand Position 1 (P1) arginine is shown, and N-terminal amino acid residues at P2 and P3 are shown in simplified format. **f** dasTMPRSS11D (left; 15 nM enzyme) and dasTMPRSS2 (right; 1.5 nM enzyme) half-maximal inhibitory concentration (IC$_{50}$) plots for the indicated small molecule and peptidomimetic inhibitors. Assays contained a final concentration of 100 μM Boc-QAR-AMC substrate and relative protease activities were determined across the first 60 s of the reaction after substrate addition. Inhibitors were pre-incubated with dasTMPRSS11D or dasTMPRSS2 for 10 minutes prior to the start of the assay. Data are shown as mean values +/− SD for experiments performed in technical duplicate across 4 independent biological replicates (total $n = 8$). Peptidomimetic 1 (PM-1): Ac-Glu-Gln-Arg-kbt. Peptidomimetic 2 (PM-2): Ac-Gln-Ser-Arg-kbt.

chromatography (Supplementary Fig. 1a). These data suggest that the dasTMPRSS11D protein underwent proteolytic cleavage to release its SEA domain, consistent with observations in HEK293 cells overexpressing full-length TMPRSS11D, where the serine protease domain was detected in conditioned media and the N-terminus containing its SEA domain remained attached to the cell surface[33]. In contrast, the eTMPRSS11D S368A protein did not undergo autocleavage activation at a protein concentration of 5 mg/mL. However, overnight incubation of eTMPRSS11D S368A with nanomolar amounts of active dasTMPRSS11D produced eTMPRSS11D S368A protein bands migrating at 25 kDa and 12 kDa when SDS-PAGE samples were reduced, and a single protein band at 40 kDa when the sample was not reduced prior to gel separation (Fig. 2d). These banding patterns suggest that the cleaved eTMPRSS11D S368A protein contained its SEA and SP domains, and the dasTMPRSS11D protease successfully activated the eTMPRSS11D S368A protein sample at its LSEQR$^{186}$ ↓ ILGG zymogen activation motif. We repeated this experiment for TMPRSS2 and found that eTMPRSS2 S441A can be cleaved at its SRQSR$^{255}$ ↓ IVGG zymogen activation motif through incubation with trace amounts of dasTMPRSS2 (Supplementary Fig. 1b, c). Thus, these overexpression studies and protease activation assays provide direct biochemical evidence that TMPRSS2 and TMPRSS11D are capable of autocleavage

activation, and that protease autocleavage activation can pose challenges in their recombinant overexpression.

### Activation motif peptides inhibit TMPRSS proteases

We developed in vitro assays to evaluate TMPRSS11D inhibitors, laying the groundwork for structure-activity relationship studies and drug discovery. For inhibitor screening, we selected the Boc-QAR-AMC substrate which had a dasTMPRSS11D $K_m$ values of 8.9 μM and Vmax of 0.10 μmol s$^{-1}$ (Supplementary Fig. 2). Since TMPRSS11D and TMPRSS2 efficiently cleave their own zymogen motifs, the C-terminus of peptides containing parts of the TMPRSS11D and TMPRSS2 zymogen motifs were derivatized with ketobenzothiazole warheads, creating peptidomimetics Ac-Glu-Gln-Arg-kbt (PM-1) and Ac-Gln-Ser-Arg-kbt (PM-2) inhibitory peptides, respectively (Fig. 2e). Both PM-1 and PM-2 were potent TMPRSS11D inhibitors with respective half-maximal inhibitory concentrations (IC$_{50}$S) of 24 [18, 31] nM and 9 [7, 11] nM (95% confidence interval reported in brackets; Fig. 2f). Nafamostat, camostat and 6-amidino-2-naphthol, which are covalent and competitive inhibitors of trypsin-like serine proteases, had weaker TMPRSS11D IC$_{50}$ values of 59 [50, 69] nM, 151 [132, 172] nM and 34 [28, 40] μM, respectively (Fig. 2f). These data suggested that TMPRSS11D was more potently inhibited by kbt-containing peptides than traditional small

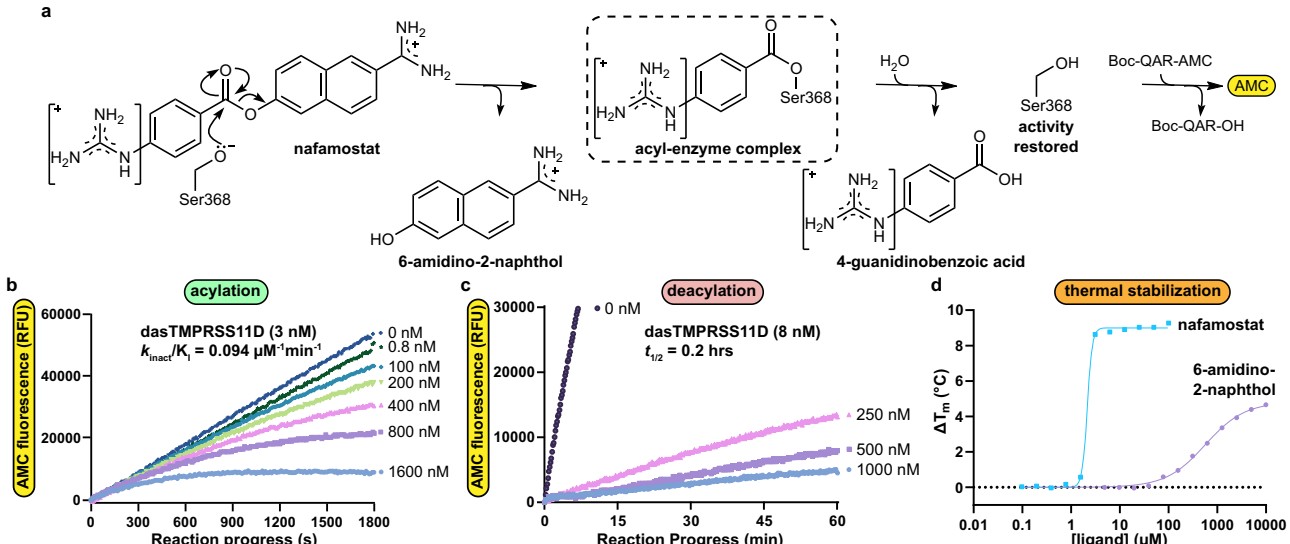

**Fig. 3 | Nafamostat rapidly acylates dasTMPRSS11D, then hydrolyzes to restore protease activity. a** The putative nafamostat covalent inhibition mechanism for TMPRSS11D. **b** Peptidase activity progress curves of dasTMPRSS11D (3 nM) with nafamostat at the indicated inhibitor concentrations added simultaneously with Boc-QAR-AMC substrate (100 μM final). **c** Peptidase activity progress curves of dasTMPRSS11D (8 nM) pre-incubated (10 min) with the indicated concentrations of nafamostat before being transferred to wells containing Boc-QAR-AMC substrate (100 μM final). Data for (**b, c**) are shown as mean values for experiments performed in technical duplicate ($n = 2$) and are consistent with results obtained across $n = 4$ independent biological replicates **d** Melting temperature shifts ($\Delta T_m$s) of dasTMPRSS11D protein in the presence of the indicated concentrations of nafamostat (teal datapoints) or 6-amidino-2-naphthol (violet datapoints) ligands. Each assay contained 2 μg dasTMPRSS11D, 5X SYPRO orange dye, and 50 mM Tris pH 8.0 with 200 mM NaCl. Data are shown as mean values for experiments performed in technical triplicate ($n = 3$), with consistent data observed across $n = 3$ independent biological replicates. The $\Delta T_m$ data were curve-fitted for one-site $EC_{50}$ in GraphPad Prism.

molecule trypsin-like serine protease inhibitors. In contrast, TMPRSS2 was inhibited by PM-1, PM-2, nafamostat and camostat with similarly potent $IC_{50}$ values of 2.2 [2.0, 2.4] nM, 2.1 [2.0, 2.3] nM, 0.80 [0.74, 0.87] nM, and 6.7 [6.0, 7.4] nM respectively, but was weakly inhibited by 6-amidino-2-naphthol with an $IC_{50}$ value of 4.0 [3.6, 4.5] μM (Fig. 2f). Our in vitro assays identified ketobenzothiazole-containing peptidomimetics as potent TMPRSS11D inhibitors, outperforming traditional small-molecule inhibitors, while TMPRSS2 inhibition remained consistent across peptides and nafamostat.

**Nafamostat acylates dasTMPRSS11D and rapidly hydrolyzes**
Nafamostat, camostat and other ester-based serine protease inhibitors rapidly acylate the conserved catalytic serine residue to block enzymatic activity, then the acyl-enzyme complex eventually hydrolyzes to restore proteolytic activity (Fig. 3a). Due to its in vitro inhibition of TMPRSS2, nafamostat and camostat were explored as COVID-19 therapeutics in clinical trials[34–36]. After hydrolysis of the acyl-enzyme complex, the product molecules have low inhibitory potency towards the protease relative to the starting ester compound[15,37]. We hypothesized that the same nafamostat inhibition mechanism applies to TMPRSS11D. To determine these parameters, nafamostat and Boc-QAR-AMC substrate were added simultaneously to wells containing dasTMPRSS11D (Fig. 3b). The reaction progress curves plateaued over time, indicating nafamostat acylates TMPRSS11D's S368 residue. We used progress curve fitting to determine kinetic microscopic rate constants and calculated a TMPRSS11D $k_{inact}/K_I$ value of 0.094 μM⁻¹min⁻¹ for nafamostat (Fig. 3b; Supplementary Fig. 3a, b). Interestingly, the nafamostat:TMPRSS11D acyl-enzyme complex rapidly hydrolyzed and restored dasTMPRSS11D peptidase activity, with an inhibition half-life ($t_{1/2}$) of only 0.2 hours (Fig. 3c; Supplementary Fig. 3c). In comparison, the TMPRSS2 $k_{inact}/K_I$ and inhibition $t_{1/2}$ values for nafamostat were previously determined to be 180 μM⁻¹min⁻¹ and 14.7 h, respectively[15]. Thus, nafamostat more potently inactivates TMPRSS2 protease activity than TMPRSS11D and

the TMPRSS2 inhibition is retained over 73-fold longer than for TMPRSS11D.

To probe differences between the TMPRSS2 and TMPRSS11D proteins that could impact nafamostat inhibition potency and inhibition $t_{1/2}$, we used differential scanning fluorimetry (DSF) to measure ligand-induced shifts in protein $T_m$s. We previously showed that 1 μM nafamostat induces a dasTMPRSS2 $T_m$ shift ($\Delta T_m$) of $25.5 \pm 0.1\,°C$ (reported as mean value ± standard deviation)[15]. Nafamostat induced dasTMPRSS11D $\Delta T_m$s from $0.10 \pm 0.05\,°C$ to $9.2 \pm 0.2\,°C$ for ligand concentrations spanning 0.1–100 μM (teal datapoints; Fig. 3d). When plotted on a semi-logarithmic scale, the $\Delta T_m$ values for dasTMPRSS11D induced by nafamostat reached saturation (teal datapoints; Fig. 3d). The data were curve-fitted to determine the half-maximal effective concentration ($EC_{50}$) for thermal stabilization, which was $2.0 \pm 0.8$ μM (teal trace; Fig. 3d). In contrast, 6-amidino-2-naphthol (dasTMPRSS11D $K_i$ value of 71 μM; Supplementary Fig. 3d) induced a maximum dasTMPRSS11D $\Delta T_m$ of $4.67 \pm 0.06\,°C$ at a compound concentration of 10 mM, and $\Delta T_m$ values did not fully saturate (violet datapoints; Fig. 3d). When dasTMPRSS11D S368A was incubated with 100 μM nafamostat, no dasTMPRSS11D S368A $\Delta T_m$s were detected (Supplementary Table 3). In contrast, 6-amidino-2-naphthol induced a maximum eTMPRSS11D S368A $\Delta T_m$ of $2.2 \pm 0.5\,°C$ (Supplementary Table 3). These biophysical data confirm that nafamostat relies on the TMPRSS11D S368 residue to induce large $\Delta T_m$ changes, likely through the formation of a 4-guanidino benzoate acyl-enzyme complex. However, nafamostat provided less thermal stabilization to TMPRSS11D compared to TMPRSS2. This difference may offer molecular insights into why the TMPRSS11D acyl-enzyme complex hydrolyzed more quickly than the TMPRSS2 acyl-enzyme complex. Overall, these kinetic and biophysical findings suggest that TMPRSS11D interacts with nafamostat like a high-affinity substrate (with a fast on-rate and slow off-rate) rather than as a potent, irreversible inhibitor of TMPRSS11D protease activity.

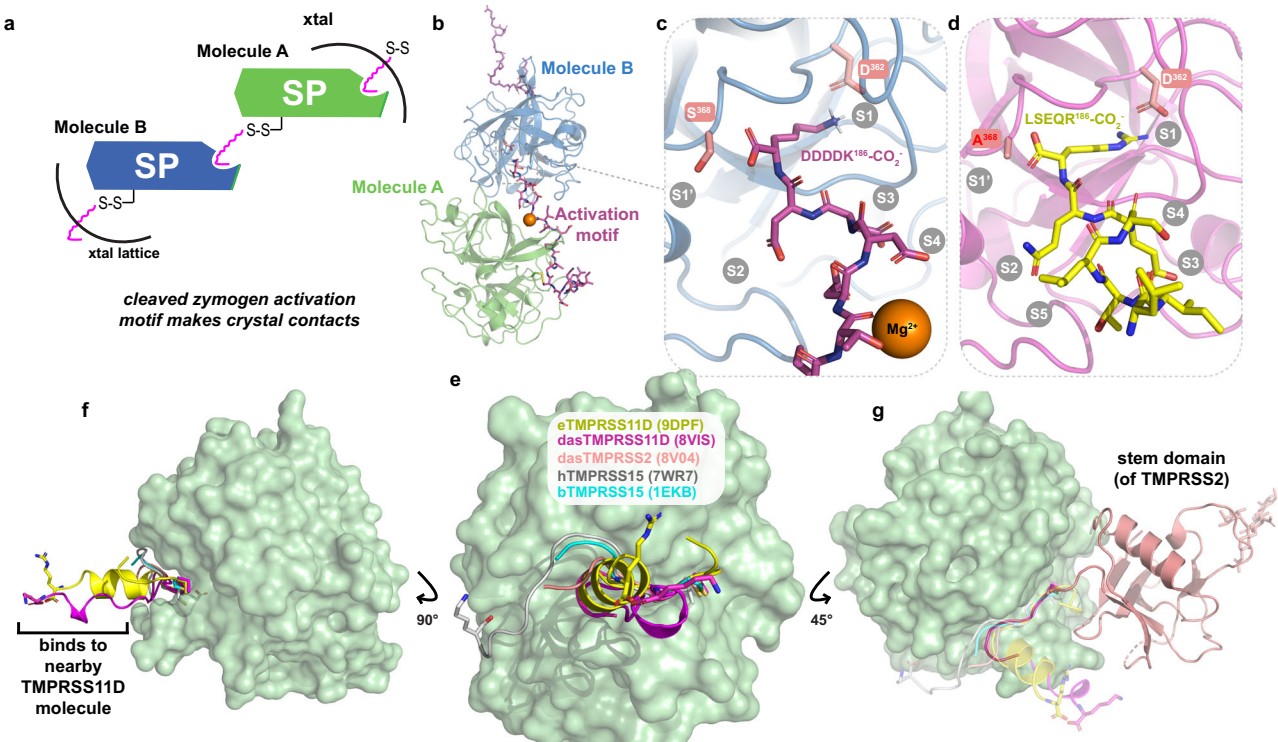

**Fig. 4 | TMPRSS11D crystallizes by using intermolecular contacts with its own zymogen motif peptide occupying the substrate binding cleft. a** Schematic of the TMPRSS11D crystal lattice containing the TMPRSS11D serine protease (SP) domain and its cleaved zymogen activation motif (magenta squiggle). **b** Cartoon representation of the dasTMPRSS11D crystal structure (PDB 8VIS). The cleaved zymogen activation motif (magenta sticks) of Molecule A interacts with the substrate binding cleft of Molecule B. **c** Zoomed-in view of the dasTMPRSS11D active site occupied by the DDDDK$^{186}$–CO$_2^-$ peptide. The TMPRSS11D catalytic S368 residue and the TMPRSS2 Subsite 1 (S1 residue) D362 are shown as salmon sticks. Additional TMPRSS11D subsites are denoted in gray text. **d** Zoomed-in view of the

eTMPRSS11D S368A active site occupied by the LSEQR$^{186}$-CO$_2^-$ peptide (PDB 9DPF). **e** Comparison of the cleaved zymogen activation motifs (attached through a disulfide bond) of the indicated TTSPs, with the SP domain of TMPRSS11D shown as a green surface. The terminal residues of the zymogen motifs of dasTMPRSS11D, eTMPRSS11D S368A, and human TMPRSS15 are shown as sticks. **f** Side view of the zymogen activation motifs from (**e**). **g** Side view of the back side of the TMPRSS11D SP domain. The stem domain of TMPRSS2 (salmon cartoon; PDB 7MEQ) is shown and is covalently attached to the SP domain of TMPRSS2 through an interdomain disulfide bond.

## Active dasTMPRSS11D cleaves the SARS-CoV-2 S protein

TMPRSS11D has been previously shown to autoactivate when overexpressed in HEK293 cells[33] and has been shown to enable in vitro SARS-CoV-2 and Influenza A infections[27,38-41]. Using recombinant SARS-CoV-2 S protein, we confirmed that active dasTMPRSS11D cleaved SARS-CoV-2 S protein as a substrate, converting the 150 kDa molecular weight protein band into proteins migrating at 100 kDa and 70 kDa, respectively (Supplementary Fig. 4). When dasTMPRSS11D was preincubated with nafamostat or 6-amidino-2-naphthol, the protease activity was blocked as indicated by the presence of the intact SARS-CoV-2 S protein band. These data match previous findings that competitive trypsin-like serine protease inhibitors can block TMPRSS11D-driven H1N1 viral entry to MDCK cells[41,42]. Our data also provided direct evidence that active TMPRSS11D can interact with the SARS-CoV-2 S protein and provided a functional assay for hit compound confirmation after high-throughput TMPRSS11D drug screening campaigns.

## TMPRSS11D cleavage motifs aid crystal packing

To better understand TMPRSS11D's esterase activity towards nafamostat, we set up cocrystallization experiments with dasTMPRSS11D preincubated with nafamostat to determine the structure of the acyl-enzyme complex. Protein crystals formed in two distinct precipitant conditions, but no electron density was observed attached to the S368 residue that was expected from the nafamostat co-structure determined for TMPRSS2[15]. Instead, the protease crystallized with the cleaved zymogen activation motif interacting with a neighboring protease molecule in the crystal lattice (magenta squiggle; Fig. 4a, b).

The dasTMPRSS11D structure was solved at a resolution of 1.59 Å and the zymogen activation motif (DDDDK$^{186}$-CO$_2^-$) was clearly resolved within the substrate binding cleft (PDB 8VIS; Fig. 4c; Supplementary Fig. 5a). We used this crystal form to next determine the structure of TMPRSS11D complexed with the more biologically relevant TMPRSS11D zymogen motif (LSEQR$^{186}$-CO$_2^-$) by crystallizing eTMPRSS11D S368A and solving its structure at 1.90 Å resolution (PDB 9DPF; Fig. 4d; Supplementary Fig. 5b). Data collection and refinement statistics are summarized in Supplementary Table 4.

Only a few structures of active TTSPs contain any electron density for the disulfide-linked, cleaved zymogen activation motif. This is because TTSP proteins purified for crystallization are typically overexpressed as inclusion bodies in *Escherichia coli* and exclusively contain the TTSP SP domain[43-45]. Notable exceptions to this protein production and crystallization trend include TMPRSS1 (hepsin; PDB 1Z8G)[46], TMPRSS2 (PDBs 7MEQ and 8V04)[15], TMPRSS13 (PDB 6KD5)[47], bovine enteropeptidase (PDB 1EKB)[48], and human enteropeptidase (PDB 7WR7)[49] which showed some electron density for their cleaved zymogen activation motifs. However, only a few zymogen motif amino acids were resolved for TMPRSS1, TMPRSS2, TMPRSS13, and bovine enteropeptidase at residues 5, 6, 6, and 7, respectively (Fig. 4e), while the cryo-EM structure of enteropeptidase (3.10 Å resolution) modeled the complete CGKKLAAQDITPK$^{186}$-CO$_2^-$ zymogen activation motif. Interestingly, both the eTMPRSS11D S368A zymogen activation motif (yellow) and dasTMPRSS11D zymogen activation motif (magenta) adopted an α-helical structure oriented perpendicular to the face of the SP domain, whereas all other TTSP zymogen motifs were β-strand

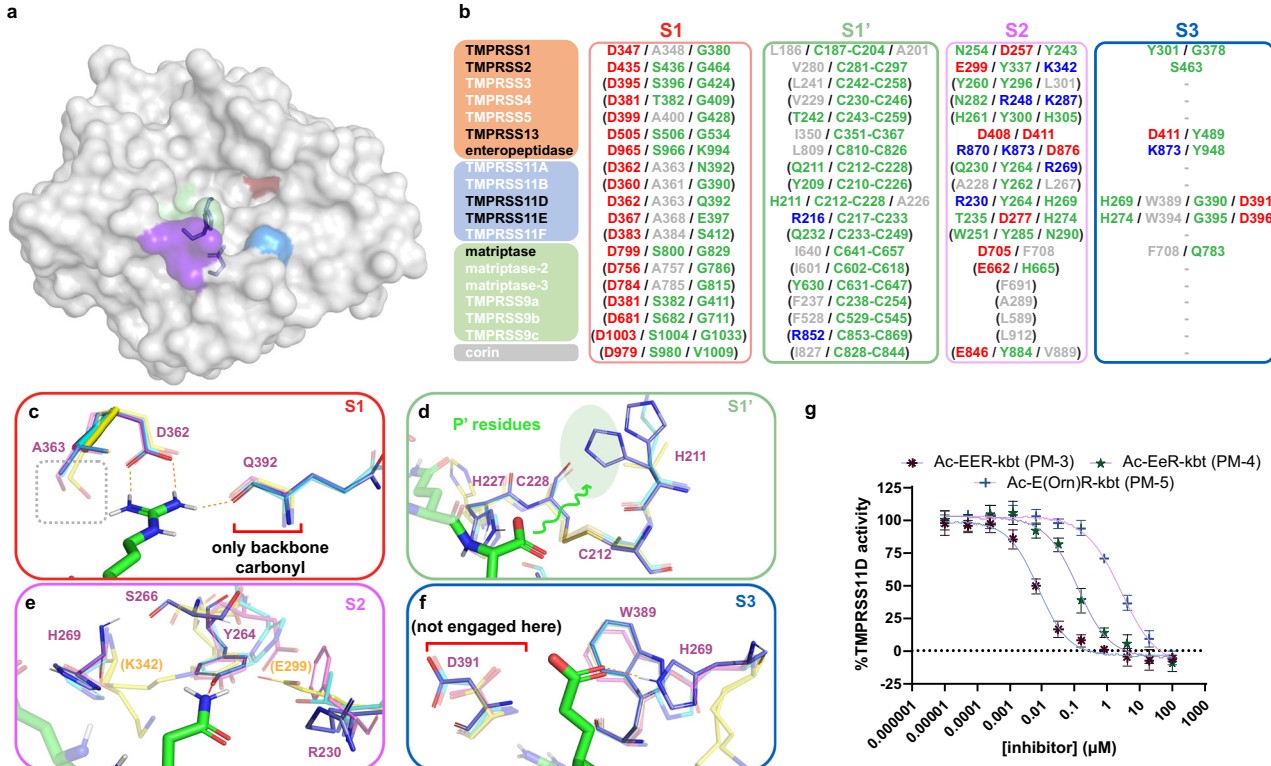

**Fig. 5 | Structural comparison of the S1', S1, S2 and S3 binding sites of TMPRSS11D, TMPRSS11E, TMPRSS2 and TMPRSS13. a** Surface representation of TMPRSS11D, with Subsite 1' (S1'; green), S1 (red), S2 (purple), and S3 (blue) subsites colored. The TMPRSS11D catalytic triad, S368-H227-D272, are shown as sticks. **b** Comparison of the amino acids at each protease subsite for every TTSP. TTSPs with experimental crystal structures are indicated in black text. Protease amino acids predicted to form salt bridges are indicated in red (electronegative) or blue (electropositive) whereas hydrophobic amino acids are indicated in gray text. Amino acids in green are predicted to participate in H-bonding with ligands. **c** S1 of TMPRSS11D shown in stick representation. The R186 ligand residue (from PDB 9DPF) is shown as green sticks. H-bonds between the ligand and the TMPRSS11D S1

are shown as dashed orange lines. TMPRSS11D residues- purple TMPRSS11E (PDB 2OQ5)-teal, TMPRSS2 (PDB 8V04)-yellow, TMPRSS13 (PDB 6KD5)-pink. **d** S1' of TMPRSS11D. The carboxylate of $R^{186}\text{-}CO_2^-$ is shown as green sticks and points towards S1' (indicated by a green squiggle). **e, f** S2 and S3 of TMPRSS11D. **g** dasTMPRSS11D (15 nM enzyme) $IC_{50}$ plot for ketobenzothiazole (kbt)-containing peptidomimetics 3-5. Assays contained a final concentration of 100 μM Boc-QAR-AMC substrate and relative protease activity was determined across the first 60 s of the reaction after substrate addition. Data are shown as mean values +/−SD for experiments performed in technical duplicate across 4 independent biological replicates (total $n = 8$). PM-3:Ac-Glu-Glu-Arg-kbt, PM-4:Ac-Glu-D-Glu-Arg-kbt, PM-5:Ac-Glu-Orn-Arg-kbt.

structures wrapped around the SP domain and/or the electronic density was poorly resolved around this region (Fig. 4e, f).

The TMPRSS11D SEA domain was not resolved in either TMPRSS11D crystal structure. To confirm that the domain was proteolytically cleaved, eTMPRSS11D S368A crystals were harvested, washed in buffer and subjected to SDS-PAGE separation and Coomassie blue staining (Supplementary Fig. 6). Only a single protein band was detected that migrated at a molecular weight of ~25 kDa which suggested that the TMPRSS11D SEA domain was not present in the protein crystal (Supplementary Fig. 6d). To predict where the SEA domain may interact with the TMPRSS11D SP domain, we superposed the TMPRSS11D crystal structure with the structure of TMPRSS2 which contained its active SP domain and SRCR domain linked by a disulfide bond (Fig. 4g). The SRCR domain of TMPRSS2, as well as other stem domains of TTSPs, supports the back face of the SP domain (salmon cartoon; Fig. 4g). This suggests that the SEA domain of TMPRSS11D may also be placed in that region prior to its proteolytic cleavage and shedding.

**TMPRSS11D S1 and S1' influence inhibitor binding**

The TMPRSS11D substrate-binding cleft consists of pockets of amino acids surrounding the Ser-His-Asp catalytic triad. These pockets–S1, S1', S2, and S3–are highlighted as colored surfaces in Fig. 5a and determine which ligand residues can bind at the corresponding P1, P1', P2, and P3 positions. To analyze these binding sites, we compared

known TTSP structures (where experimental data are available) and predicted models (based on homology modeling and sequence alignments), identifying key amino acids in the S1, S1', S2, and S3 pockets (Fig. 5b).

The TMPRSS11D S1 is formed by D362 and Q392 that mediate salt bridges and H-bonds, respectively, with P1 Arg or Lys residues (Fig. 5c). Unlike TMPRSS2 and TMPRSS13, the TMPRSS11D S1 contains A363 instead of a Ser residue that can H-bond with the guanidine or amine of P1 Arg or Lys residues. Furthermore, a Gly residue is typically present at the equivalent position of the TMPRSS11D Q392 residue for most other TTSPs, excepting enteropeptidase (Lys), TMPRSS11E (Glu), TMPRSS11A (Asn), and corin (Val; Fig. 5b). Thus, TMPRSS11D's lack of the conserved Ser and Gly residues within S1 may explain why nafamostat and 6-amidino-2-naphthol did not induce as large of a $\Delta T_m$ for TMPRSS11D as for TMPRSS2. Furthermore, the fewer molecular contacts made by the nafamostat acyl-enzyme complex could lead to the rapid hydrolysis that is observed for TMPRSS11D but not TMPRSS2.

The S1' of TMPRSS11D (which recognizes the kbt portion of peptidomimetic inhibitors) includes the conserved C212-C228 disulfide found in every TTSP (Fig. 5d). However, TMPRSS11D has a H211 residue whereas TMPRSS11E employed R216, TMPRSS13 used I350, and TMPRSS2 used V280, at their S1'. These subtle differences may confer some distinct P1' residue preference to TMPRSS11D compared to these other proteases. Although not investigated in the current study, S1' differences may present an opportunity to further modify the kbt

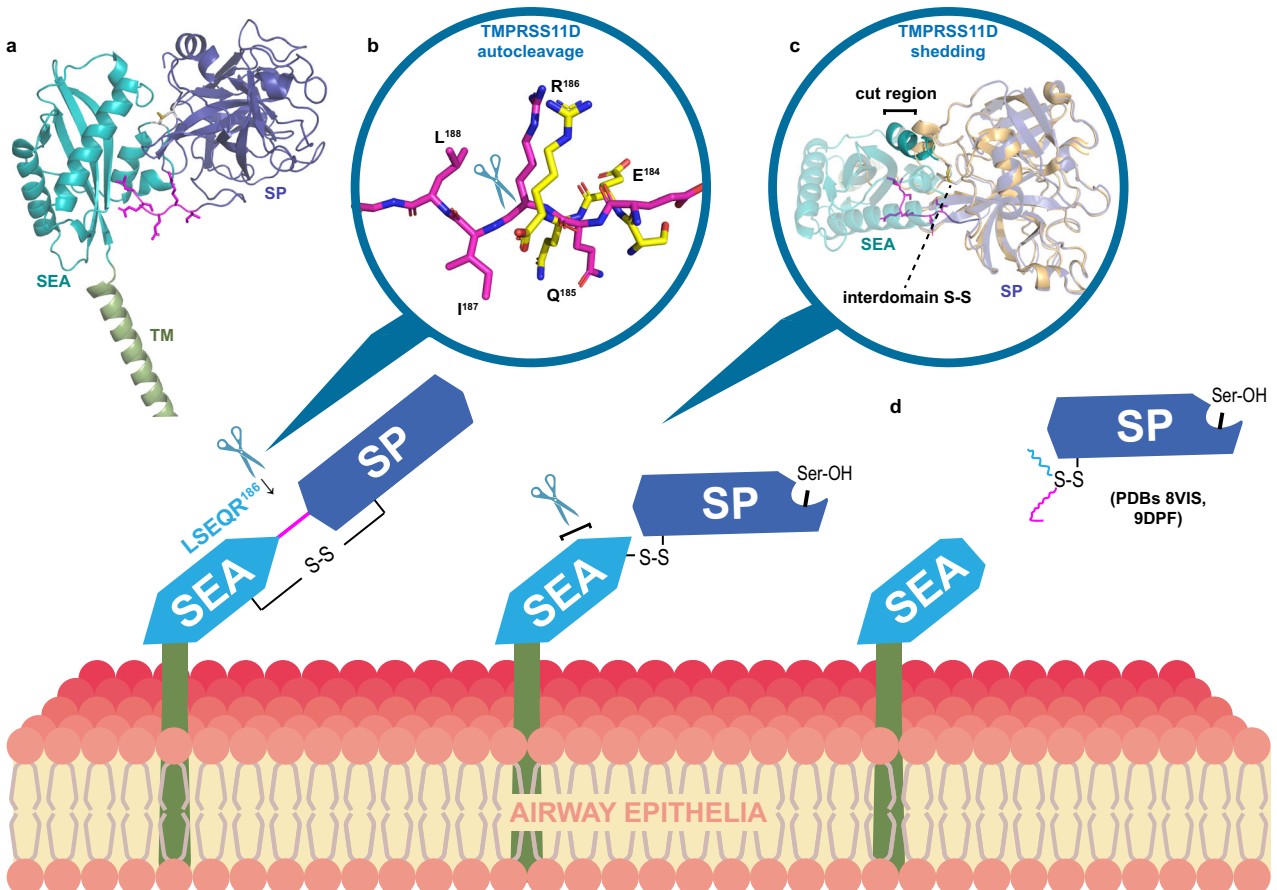

**Fig. 6 | A proposed model of TMPRSS11D autocleavage activation and shedding from the cell surface. a** AlphaFold 2.0 model of full-length human TMPRSS11D (AF-O60235-F1). The transmembrane (TM) domain (green), the Sea urchin, Enteropeptidase and Agrin (SEA) domain (teal), and the serine protease (SP) domain (purple) are shown in cartoon representation. The TMPRSS11D zymogen activation motif is shown in magenta sticks. **b** Superposed model of the AlphaFold 2.0 TMPRSS11D protein structure (magenta sticks) and the cleaved TMPRSS11D zymogen activation motif (PDB 9DPF; yellow sticks). The TMPRSS11D cleavage site spanning the R186-I187 peptide bond is denoted with a scissor graphic. **c** Cartoon representation of the TMPRSS11D cleavage event in the SEA domain leading to protease shedding from the cell surface. The SP domains of the AlphaFold 2.0 TMPRSS11D structure (purple) and the dasTMPRSS11D crystal structure (cartoon) were superposed. The suspected SEA domain cleavage site is indicated with a black bar. **d** Graphical representation of the shed TMPRSS11D SP domain.

portion of peptidomimetic inhibitors (P1′) to selectively engage TTSPs, similar to how a carboxylate modification in the kbt scaffold has been used to enhance specificity for human thrombin by interacting with its K60 residue[50].

## The TMPRSS11D and TMPRSS2 S2 are distinct and targetable
The S2 of TMPRSS11D contains a combination of R230, Y264, and H269 which is distinct from every other TTSP whereas the TMPRSS11D S3 closely resembled the TMPRSS11E S3 (Fig. 5b, e, f). We therefore focused our attention on identifying favorable ligand interactions with this site. TMPRSS2 also contains a positively charged amino acid within its S2, K342 as well as a negatively charged E299 at the equivalent position of TMPRSS11D's R230. Notably, R230 did not make direct interactions with a P2 Gln or P2 Asp in our TMPRSS11D structures (Fig. 5e), suggesting that improvements could be made by modifying the P2 amino acid. In our biochemical assays, peptidomimetics PM-1 and PM-2 containing P2 Gln and Ser, respectively, were tolerated by the S2 of TMPRSS11D and TMPRSS2. Similarly, substrate peptide Boc-QAR-AMC containing a P2 Ala was efficiently cleaved by TMPRSS11D and TMPRSS2 which indicated a P2 Ala residue is tolerated by both proteases.

To probe if TMPRSS11D's R230 residue could be pharmacologically targeted, we prepared three analogs of PM-1 with P2 substitutions, Ac-Glu-Glu-Arg-kbt (Ac-EER-kbt; PM-3), Ac-Glu-D-Glu-Arg-kbt (Ac-EeR-kbt; PM-4), and an ornithine (Orn)-containing peptidomimetic Ac-Glu-Orn-Arg-kbt (PM-5). PM-3 was the most potent TMPRSS11D inhibitor identified with an $IC_{50}$ of 7.1 [6.0, 8.4] nM (Fig. 5g). PM-4 with a P2 D-Glu had moderate TMPRSS11D binding affinity with an $IC_{50}$ of 111 [92, 134] nM (Fig. 5g). PM-5 containing a positively charged P2 Orn lost substantial TMPRSS11D binding affinity relative to all other peptidomimetics, with a TMPRSS11D $IC_{50}$ value of 2.1 [1.8, 2.4] μM (Fig. 5g). For TMPRSS2, PM-3 and PM-5 retained potent binding affinity with TMPRSS2 $IC_{50}$ values of 3.2 [2.9, 3.4] nM and 6.2 [5.5, 7.0] nM, respectively, whereas PM-4 had moderate binding affinity with an $IC_{50}$ value of 42 [38, 48] nM (Supplementary Fig. 7). These results highlight the importance of P2 residue selection in modulating inhibitor potency and selectivity, with PM-3 emerging as a highly potent dual TMPRSS11D and TMPRSS2 inhibitor, likely due to favorable electrostatic interactions. In contrast, PM-5 exhibited selectivity for TMPRSS2 over TMPRSS11D, suggesting that electrostatic repulsion at TMPRSS11D's R230 contributes to its reduced binding affinity.

## Modeling TMPRSS11D zymogen activation and shedding
The TMPRSS11D SEA domain was not resolved in our TMPRSS11D crystal structure, motivating us to investigate modeled structures to develop a structure-informed understanding of full-length TMPRSS11D zymogen activation and shedding from the cell surface. The AlphaFold2 TMPRSS11D structure (AF-O60235-F1)[51,52], which depicts the

zymogen form of the protease, modeled the zymogen activation motif in a similar orientation as that of the pro-matriptase structure (Fig. 6a; Supplementary Fig. 8a). As expected, the SEA domain of the Alpha-fold2 TMPRSS11D structure superposed closely with the 1.92 Å resolution crystal structure of the SEA domain of mouse TMPRSS11D (PDB 2E7V; Supplementary Fig. 8b). Furthermore, the placement of the SEA domain relative to the TMPRSS11D SP domain was similar to the TMPRSS2 crystal structure containing its SP and SRCR domains, but the SEA domain was larger and extended in the opposite direction of the SRCR domain (Supplementary Fig. 8c). Thus, the AlphaFold2 TMPRSS11D structure serves as a plausible prediction of the zymogen form of the full-length protease.

As the nascent, membrane-bound TMPRSS11D zymogen is trafficked along the secretory pathway and reaches the cell surface, it is possible that an active TMPRSS11D protease molecule (or another trypsin-like serine protease) recognizes the zymogen motif as a substrate and cleaves the R186-I187 peptide bond before or after it reaches the cell surface. Accordingly, we investigated if the conformation of the zymogen motif was appropriate for cleavage by TMPRSS11D (Fig. 6b). We superposed the cleaved zymogen motif of the eTMPRSS11D S368A structure (LSEQR$^{186}$-CO$_2^-$; yellow sticks; Fig. 6b) upon the intact, exposed zymogen motif of the AlphaFold2 TMPRSS11D structure (magenta sticks; Fig. 6b). The conformations of the ligands were similar and indicated that an active TMPRSS11D molecule could potentially bind and cleave this exposed zymogen motif depicted in the AlphaFold2 structure. After the TMPRSS11D molecule has undergone zymogen activation, it can exert its proteolytic function at the cell surface. Our model also is consistent with subsequent cleavage by TMPRSS11D of a residue near the C-terminus of the SEA domain (bolded teal cartoon; Fig. 6c) thereby enabling shedding of the active TMPRSS11D SP domain from the cell surface (Fig. 6d).

## Discussion

The zymogen activation step of TTSPs is a critical aspect of their biology and determines when and where the TTSP harbors its matured proteolytic activity. Each TTSP family member has a distinct sequence of amino acids within their zymogen activation motifs. It remains unclear which proteases recognize and cleave which TTSP zymogen activation motif, and whether TTSP cleavage activation occurs during or after trafficking of the protease to the cell surface. The biochemical and structural data presented here showed that TMPRSS11D and TMPRSS2 recognized their own (and each other's) zymogen activation motifs, allowing them to turn on their own and potentially each other's proteolytic activity. Although evidence suggests that these proteases undergo autocleavage activation in cells[19,33,53], we have not yet evaluated their potential to facilitate cross-activation in cell-based systems in this study. To facilitate the analysis of soluble forms of these TTSPs, we replaced their zymogen activation motifs with DDDDK; this delayed their zymogen activation and enabled protein overexpression, biochemical assays, and protein crystallization. We confirmed that the DDDDK sequence does not interfere with protease activity assays for high-throughput drug screening applications. Furthermore, these proteases efficiently cleave the SARS-CoV-2 S protein in functional protease activity assays, confirming they have their expected biological activity towards important respiratory virus entry proteins[24,39].

Nafamostat mesylate and the structurally related molecule camostat have been proposed as antiviral agents disabling TTSPs and have repeatedly been used in cell studies to block TMPRSS2/11D-mediated viral entry at the cell surface[3,5,54,55]. Our kinetic data indicated that TMPRSS11D interacted with nafamostat as a substrate, rapidly forming an acyl-enzyme complex and then hydrolyzing the ester to convert it to the product molecules 4-guanidino benzoic acid and 6-amidino-2-naphthol (Fig. 3). 6-amidino-2-naphthol inhibited TMPRSS11D, but with much weaker potency than its parent molecule

nafamostat. Thus, TMPRSS11D may not be disabled by nafamostat (and other related esters such as camostat) over a prolonged period. TMPRSS11D's esterase activity suggested that ester-based compounds may not be effective as antivirals for human airway cells expressing this enzyme. When developing ester drugs targeting TTSPs, the rate at which ester compounds are broken down should be kinetically characterized[15,56]. Additionally, an ester that is a potent inhibitor of a particular TTSP target could be rapidly turned over by another co-expressed TTSP and lead to rapid elimination of the ester. In contrast, ketobenzothiazole-containing peptidomimetics PM-1 and PM-2 which were based on the zymogen activation motif sequences of TMPRSS11D and TMPRSS2, respectively, were nanomolar potency TMPRSS11D inhibitors that were not irreversibly cleaved during prolonged incubation with the protease.

The TMPRSS11D crystal structure provided molecular insights into the reversibility of nafamostat inhibition. The TMPRSS11D S1 lacks an H-bond donor relative to TMPRSS2's S1, and nafamostat cannot thermally stabilize (measured through nafamostat-induced $\Delta T_m$s) TMPRSS11D at the same magnitude as TMPRSS2[15]. All other HAT/DESC TTSP subfamily members lack this H-bond donor residue, as they all contain Ala residues at this position in their S1 (Fig. 5b). Other electrostatic and steric features of the substrate binding cleft may promote hydrolysis of the acyl-enzyme complex. Overall, the protease activity of the HAT/DESC subfamily of TTSPs may not be as effectively disabled by nafamostat as other TTSP subfamilies.

The dasTMPRSS11D crystallization trials were initially performed in the presence of high concentrations of nafamostat or 6-amidino-2-naphthol. Protein crystals formed in the presence of each ligand, but the final crystal structures contained no electron density matching the ligands. Instead, the zymogen activation motif of dasTMPRSS11D, DDDDK$^{186}$-CO$_2^-$, was found occupying the substrate binding cleft of a nearby TMPRSS11D molecule within the crystal lattice. We repeated this crystallization strategy for TMPRSS11D with its native zymogen activation motif, LSEQR$^{186}$-CO$_2^-$, and determined the crystal structure of a TTSP interacting with the N-terminal (non-prime) component of its own zymogen activation motif. This structure explained why dasTMPRSS11D was capable of cleaving and activating eTMPRSS11D S368A and explained why TMPRSS11D can autoactivate when overexpressed in HEK293 cells[33].

Using the eTMPRSS11D S368A crystal structure complexed with the LSEQR-CO$_2^-$ peptide ligand, we mapped the S1'-S3 subsites of TMPRSS11D, providing an opportunity to rationally target the TMPRSS11D binding cleft with peptidomimetic inhibitors, focusing on exploring S2 interactions in the current study. The S2 of TMPRSS11D contains a combination of R230, Y264, and H269, distinct from every other TTSP. The S2 of TMPRSS2, by contrast, contains E299, Y337, and K342 at the equivalent positions. Both proteases accommodated a P2 Gln residue, as evidenced by potent inhibition by PM-1, but showed weaker affinity than PM-2, which contained a P2 Ser residue. Analogs PM-3, PM-4, and PM-5 explored the effects of incorporating P2 Glu, D-Glu, and Orn residues on TMPRSS11D and TMPRSS2 binding affinity. A P2 Glu yielded the most potent reported TMPRSS11D inhibitor, PM-3, with an IC$_{50}$ of 7.1 [6.0, 8.4] nM (95% confidence interval reported in brackets). PM-3 was also the most potent TMPRSS2 inhibitor in this study, with an IC$_{50}$ of 3.2 [2.9, 3.4] nM. In contrast, introducing a positively charged Orn (PM-5) selectively inhibited TMPRSS2 with an IC$_{50}$ of 6.2 [5.5, 7.0] nM but substantially reduced TMPRSS11D binding affinity (IC$_{50}$ = 2.1 [1.8, 2.4] μM). PM-4, containing a P2 D-Glu, retained moderate binding affinity for both TMPRSS11D (IC$_{50}$ = 111 [92, 134] nM) and TMPRSS2 (IC$_{50}$ = 42 [37, 48] nM). This moderate binding affinity, despite the introduction of a D-amino acid, could be explained by continued salt bridge and/or H-bond interactions with the S2 residues of TMPRSS11D and TMPRSS2 but induced a new conformation of the P3 ligand residue. We therefore identified a potent dual TMPRSS11D and TMPRSS2 inhibitor (PM-3), a TMPRSS2-selective inhibitor (PM-5),

and a D-amino acid–containing peptidomimetic with moderate affinity for both proteases (PM-4). Unlike TMPRSS11D, TMPRSS2 has been widely targeted with peptidomimetic inhibitors[6,7,57], but to our knowledge, P2 Glu, D-Glu, or Orn have not previously been explored as scaffolds for TMPRSS2 inhibition. Further structure-guided optimization of PM-3-5, particularly through substitution of the P3 ligand residue, may enhance potency and selectivity towards TMPRSS11D and/or TMPRSS2 by enhancing complementarity with their substrate binding clefts.

The protease crystallization strategy identified here may be generally applicable to TTSPs. Notably, other TTSPs have been structurally characterized with their disulfide-linked zymogen activation motifs placed in the same region as our crystal structures at the back of the TTSP SP domain (Fig. 4). However, the TTSP stem domains may sterically block the crystallization tag from interacting with a neighboring protease molecule in the crystal lattice. To test this, we overexpressed and purified dasTMPRSS11D with a S368A mutation and carefully cleaved the protease through treatment with enteropeptidase (Supplementary Fig. 9). The protease was fully cleaved by enteropeptidase treatment and had the same DDDDK-$CO_2^-$ peptide available to act as a crystallization tag as the original dasTMPRSS11D crystal structure. However, the SDS-PAGE gel migration patterns of this sample suggested that the protein contained the SEA domain (Supplementary Fig. 9b) so the crystallization experiment contained the complete TMPRSS11D ectodomain. No protein crystals formed under the same precipitant screening conditions as dasTMPRSS11D and eTMPRSS11D S368A. Interestingly, the SEA domain has previously been reported to impede both protein crystallization and structure determination by cryo-EM[49]. Thus, to reapply this crystallization strategy to other TTSPs, the stem domains of the protease may need to be removed to enable access to the crystallization tag by neighboring protease molecules, with recombinant protein construct designs informed by the TMPRSS11D crystal structures here.

The biochemical and structural data outlined here have provided an improved understanding of TMPRSS11D and TMPRSS2 zymogen cleavage activation and explained how these proteases may be capable of autocleavage activation in human cells. Thus, TMPRSS2 and TMPRSS11D are two TTSPs that rapidly autoactivate in vitro and are efficient drivers of SARS-CoV-2 infection in vitro and in vivo[7]. TTSP autocleavage activation may therefore be an indicator of a protease's ability to drive respiratory virus infections, and future work may help define how TTSP zymogen activation influences viral pathobiology and help prioritize human protease targets for antiviral development.

## Methods

### Chemicals and biochemicals

All biochemical reagents and solvents were purchased from Sigma-Aldrich (Oakville, ON, Canada) unless indicated otherwise. The 2-chlorotrityl chloride resin, with a loading capacity of 1.2 mmol/g, was purchased from Matrix Innovation (Quebec, QC, Canada). All commercially available Fmoc-protected amino acids and coupling reagents were purchased from Combi Blocks (San Diego, CA, USA), Chem-Impex (Wood Dale, IL, USA), or Matrix Innovation (Quebec, QC, Canada) at the highest purity available. Trifluoroacetic acid (TFA) and N,N-Diisopropylethylamine (DIPEA) were purchased from Chem-Impex (Wood Dale, IL, USA). N,N-Dimethylformamide (DMF) and diethyl ether ($Et_2O$) were purchased from Fisher Scientific (Hampton, NH, USA).

### Construct design and cloning

TMPRSS11D cDNA (nucleotide accession # BC125196) was purchased from Transomic and constructs encoding the soluble TMPRSS11D ectodomain spanning residues 44-418 were subcloned into the pFHMSP-LIC C or pFHMSPN-avi-TEV-LIC baculovirus donor vectors. All constructs contained a N-terminal honeybee melittin signal sequence

peptide. For pFHMSP-LIC C, proteins had a C-terminal $His_8$ tag. For pFHMSPN-avi-TEV-LIC, proteins had a N-terminal $His_6$ tag followed by an Avi tag for biotinylation, then a TEV cleavage site. All mutagenesis primers are listed in Supplementary Table 1 and mutagenesis was achieved using the LIC method. For the engineered active TMPRSS11D construct (dasTMPRSS11D), L182D/S183D/E184D/Q185D/R186K mutations were implemented. The donor vectors containing the engineered TMPRSS11D gene were transformed into *Escherichia coli* DH10Bac cells (Thermo Fisher; Cat# 10361012) to generate recombinant viral bacmid DNA. Gibco™ Sf9 cells (Thermo Fisher Cat# 12659017) were transfected with Bacmid DNA using JetPrime transfection reagents (Poly-Plus Transfection Inc.; Cat# 114-01) according to the manufacturer's instructions, and recombinant baculovirus particles were obtained and amplified from P1 to P2 viral stocks. P1 viral stocks were used for protein test expression studies with suspension culture of baculovirus-infected insect cells. For scaled-up productions of TMPRSS11D proteins, recombinant P2 viruses were used.

The SARS-CoV-2 Spike ectodomain HexaPro construct was a gift from J. McLellan, and the S1/S2 site was restored (GSAS685->RRAR; HexaFurin construct) through site-directed mutagenesis as previously described[15].

### Baculovirus-mediated protein production in Sf9 cells

Sf9 cells were grown in I-Max Insect Medium (Wisent Biocenter; Cat# 301-045-LL) to a density of $4 \times 10^6$ cells/mL and infected with 20 mL/L of suspension culture of baculovirus-infected insect cells prior to incubation on an orbital shaker (145 rpm, 26 °C).

### Recombinant TMPRSS11D protein purifications

TMPRSS11D ectodomain proteins were produced through secreted expression and purified using a similar protocol to TMPRSS2[15]. Cell culture medium containing the final secreted protein products AA-[TMPRSS11D (44-418)]-EFVEHHHHHHH (for dasTMPRSS11D) or AAPEMHHHHHHEFMSGLNDIFEAQKIEWHEGSAGGSGENLYFQG-[TMPRSS11D (44-418) (for eTMPRSS11D S368A and dasTMPRSS11D S368A) were collected by centrifugation (20 min, 10 °C, 6000 × g) 4-5 days post-infection when cell viability dropped to 55–60%. Media was adjusted to pH 7.4 by addition of concentrated PBS stock, then supplemented with 15 mL/L settled $Ni^{2+}$-NTA resin (Qiagen) at a scale of 3 L and distributed at a scale of 1.5 L to 2.8 L glass flasks. Flasks were shaken for 1 h at 16 °C (110 rpm), then bead-media mixtures were transferred to 0.5 L gravity flow columns (Bio-Rad). Beads were washed with 3 column volumes (CVs) PBS prior to elution with 1.5 resin bed volumes of Elution Buffer (PBS supplemented with 250 mM imidazole). Crude protein was concentrated using 10 kDa MWCO Amicon filters and washed into PBS to remove excess imidazole. Protein samples were prepared for SDS-PAGE with either reducing (5 mM 2-mercaptoethanol) Laemelli dye and thermally denatured for 5 minutes, or nonreducing Laemelli dye and were not thermally denatured. Concentrated $Ni^{2+}$-NTA IMAC elution samples were passed through a 0.22 µm syringe filter and injected to a Superdex75 gel filtration column pre-equilibrated with Size-Exclusion Chromatography (SEC) Buffer (50 mM Tris pH 8.0, 200 mM NaCl). SEC fractions containing TMPRSS11D proteins were pooled, concentrated to 5 mg/mL, then flash frozen in liquid nitrogen and stored as aliquots at −80 °C in advance of assays.

### Production of peptidomimetic compounds

Arginine-ketobenzothiazole tripeptides were synthesized using a combination of solution and solid-phase syntheses according to methods previously described for TMPRSS6 inhibitors[58]. The crude peptidomimetics were purified over a reverse-phase ACMP-10-25030P preparative HPLC column (Waters). The purity of the peptidomimetic compounds was confirmed by UPLC-MS (Acquity UPLC® CSHTM C18 (2.1 × 50.0 mm) column). Only the predominant diastereoisomer was isolated with a purity exceeding 95% and used in subsequent

experiments. The identity of each peptidomimetic was confirmed by high-resolution mass spectrometry on a maXis 3G orthogonal mass spectrometer (ESI-QqTOFMS) (Bruker Daltonik; Bremen, Germany) using electrospray ionization in positive (or negative) ion mode. Each mass spectrometry analysis was conducted once for each peptidomimetic ($n = 1$; 5 peptidomimetics total). The calculated and detected masses, as well as the relative purity for each peptidomimetic, are summarized in Supplementary Table 2 and additional synthesis and purification details are available in Supplementary Methods. LC-MS chromatograms and MS spectra are provided in Supplementary Figs. 10–19.

## TMPRSS11D crystallization and data collection

SEC-purified dasTMPRSS11D protein (3 mg/mL) was incubated with excess nafamostat (5:1 compound:protease stoichiometry) or excess 6-amidino-2-naphthol (100:1 compound:protease stoichiometry) at 4 °C for 15 min, then concentrated to 30 mg/mL using a 10 kDa MWCO Amicon filter and centrifuged (14,000 rpm, 10 min, 4 °C). SEC-purified eTMPRSS11D S368A protein (at 20 mg/mL) was similarly prepared for crystallization trials but was not incubated with nafamostat or 6-amidino-2-naphthol. Protein samples were subjected to automated screening at 18 °C in 96-well Intelliplates (Art Robin) using the Phoenix protein crystallization dispenser (Art Robbins). Protein was dispensed as 0.5 μL sitting drops and mixed 1:1 with precipitant. The RedWing and SGC precipitant screens were tested. A single, large crystal was obtained for the nafamostat:dasTMPRSS11D sample with precipitant solution containing 20% PEG1500, 0.2 M MgCl₂, and 0.1 M HEPES pH 7.5. The 6-amidino-2-naphthol:dasTMPRSS11D sample produced crystals with precipitant condition containing 20% PEG3350 and 0.2 M Mg(NO₃)₂. For eTMPRSS11D S368A, protein crystals were obtained as sitting drops grown over precipitant solution containing 0.5 M C₂H₂MgO₄ and 0.1 M Bis-Tris pH 6.5. To acquire diffraction-quality eTMPRSS11D S368A protein crystals, drops were reset as 2 μL hanging drops on glass slides. All diffraction-quality TMPRSS11D protein crystals were cryo-protected with reservoir solution containing 10% ($v/v$) ethylene glycol and cryo-cooled in liquid nitrogen. X-ray diffraction data were collected on the CMCF-ID beamline at the Canadian Light Source with a Dectris Eiger X 16 M detector.

## Solving the TMPRSS11D crystal structures

X-ray diffraction data were processed with HKL-3000[59]. Initial phases for the dasTMPRSS11D crystal structure were obtained by molecular replacement using Phaser MR[60] with the catalytic chain of TMPRSS11E (PDB 2OQ5) as a search model. The dasTMPRSS11D crystal space group was P4₃2₁2 and 4 TMPRSS11D SP domain protein molecules related by translational non-crystallographic symmetry were observed in the asymmetric unit. Model building was performed in COOT and refined with REFMAC5.8.0352[61]. The dasTMPRSS11D stem chain residues 166–186 were manually built into electron density within the TMPRSS11D substrate binding cleft. For eTMPRSS11D S368A, the crystal space group was P2₁2₁2₁ and the structure was solved using molecular replacement with the dasTMPRSS11D crystal structure. For eTMPRSS11D S368A, two TMPRSS11D SP domain protein molecules were found in the asymmetric unit. Both structures were validated by Molprobity[62]. The dasTMPRSS11D crystal structure (1.59 Å) and eTMPRSS11D S368A crystal structure (1.90 Å) were uploaded to the PDB under accession codes 8VIS and 9DPF, respectively.

## Protein sequence alignments and structure superpositions

The FASTA sequences of human TTSP family members were accessed through Uniprot (isoform 1) and aligned using Clustal Omega[63] and annotated with ESPript v.3.0[64]. Protein structures were accessed from the PDB and superposed using PyMOL v2.5.7. Protein structure figures were prepared in PyMOL v2.5.7.

## TMPRSS11D peptidase assays and IC₅₀ determination

dasTMPRSS11D peptidase assays with fluorogenic Boc-Gln-Ala-Arg-AMC substrate (Bachem Cat # 4017019.0025) were carried out using a similar protocol to dasTMPRSS2[15]. Assays were carried out at a scale of 50 μL in Greiner Black 384-well microplates, with AMC fluorescence monitored on a BioTek Synergy H1 plate reader (Gen5 v3.03 software package) at 341 nm:441 nm excitation: emission. TMPRSS11D half-maximal inhibitory concentration (IC₅₀) assays contained a final concentration of 15 nM dasTMPRSS11D in Assay Buffer (25 mM Tris pH 8.0, 75 mM NaCl, and 2 mM CaCl₂). TMPRSS2 IC₅₀s were determined with a final concentration of 1.5 nM dasTMPRSS2 in the same buffer as dasTMPRSS11D. Protein concentration was initially estimated using absorbance at 280 nm (A₂₈₀) with molar extinction coefficients of 52,940 M⁻¹cm⁻¹ and 99,350 M⁻¹cm⁻¹ for the catalytic domain of dasTMPRSS11D and the ectodomain of dasTMPRSS2, respectively. For dasTMPRSS2, the enzyme concentration in the assay was determined as 2-fold the nafamostat IC₅₀ value due to its exceptional inhibitory potency and utility as a TMPRSS2 burst titrant. For dasTMPRSS11D, the enzyme concentration in the assay was determined through reaction progress curve fitting for various dasTMPRSS11D enzyme dilutions incubated with saturating concentrations of Boc-QAR-AMC substrate (100 μM; Boc-QAR-AMC Km = 8 μM), with curve-fitted data available in Supplementary Fig. 3a.

Nafamostat mesylate (MedChemExpress Cat# HY-B0190A), camostat mesylate (MedChemExpress Cat# HY-13512), 6-amidino-2-naphthol methanesulfonate (TCI Cat# A1193), and peptidomimetics 1–5 were prepared as fresh DMSO stocks immediately prior to inhibition assays. To determine dasTMPRSS11D and dasTMPRSS2 IC₅₀s, all compounds were tested across a 100 μM – 10 pM concentration range (log₅ dilution series; 11 compound concentrations) except for 6-amidino-2-naphthol which was tested across a 1000 μM – 100 pM concentration range (log₅ dilution series; 11 compound concentrations). DMSO was used as a vector control. Compounds were transferred to wells containing dasTMPRSS11D or dasTMPRSS2 and incubated for 10 min. Boc-QAR-AMC substrate was transferred to wells (100 μM final concentration) and plates immediately read for AMC fluorescence. Initial reaction velocity slopes were tabulated across the first 0–120 s of the assay and normalized (as a percentage) relative to uninhibited enzyme (DMSO control). Inhibition data was generated in technical duplicate and repeated across 4 independent biological replicates (total $n = 8$). All 8 compounds in the study were tested simultaneously (on the same microplates) for dasTMPRSS11D or dasTMPRSS2 to enable comparisons of their IC₅₀s within the same assay. Normalized reaction velocity data were plotted and curve-fitted using the Absolute IC₅₀ function in Graphpad Prism and IC₅₀ values were reported as mean values and 95% confidence intervals.

## TMPRSS11D inhibitor kinetic parameter determination

The dasTMPRSS11D covalent inactivation parameter $k_{inact}/K_I$ and noncovalent, competitive inhibition constant ($K_i$) were determined using inhibitor coaddition assays described previously[15] and reaction progress curves were curve-fitted using DynaFit 4.0, with DynaFit scripts available in Supplementary Methods. Stocks of Boc-QAR-AMC substrate with various concentrations of nafamostat (final concentrations of 1600, 800, 400, 200, 100, and 0.8 nM) or 6-amidino-2-naphthol (final concentrations of 500, 250, 125, 62.5, 31.3, 16, and 1 μM) were transferred to wells containing dasTMPRSS11D enzyme (3 nM final) using an Agilent Bravo liquid-liquid transfer device and AMC fluorescence was immediately read. Reaction progress was monitored for 30 min. The simplified Boc-QAR-AMC substrate conversion rate, $k_{sub}$, was determined by curve fitting reaction progress fluorescence data in the presence of varying concentrations of

dasTMPRSS11D enzyme using Model 1,

$$E + S \rightarrow E + P: \quad k_{sub} \qquad (1)$$

with $k_{sub} = 0.049 \, \mu M^{-1}s^{-1}$ set as a fixed parameter in subsequent kinetic analyses. To determine nafamostat's dasTMPRSS11D covalent inactivation rate, reaction progress curves were fitted to determine the combined rate constant $k_{inact}/K_I$ using Model 2,

$$E + I \rightarrow E - I: \quad k_{inact}/K_I \qquad (2)$$

which assumed a one-step kinetic mechanism and $k_{inact}/K_I$ in units of $\mu M^{-1}s^{-1}$. The nafamostat inhibition half-life ($t_{1/2}$) value was determined by pre-incubating 8 nM dasTMPRSS11D with various concentrations of nafamostat (1000, 500, and 250 nM) for 3 minutes prior to transfer to wells containing Boc-QAR-AMC substrate. Reaction progress was monitored by fluorescence for 3 h and the data was curve-fitted to determine the nafamostat hydrolysis rate ($k_{hydrolysis}$) and nafamostat inhibition $t_{1/2}$ value using Model 3,

$$E - I \rightarrow E + I': \quad k_{hydrolysis}$$
$$t_{1/2} = \ln(2)/k_{hydrolysis} \qquad (3)$$

To determine the dasTMPRSS11D $K_i$ for 6-amidino-2-naphthol, reaction progress curves were fitted to determine the microscopic rate constant $k_{dI}$ (Model 4),

$$E + I \rightarrow E.I: \qquad k_{aI}$$
$$E.I \rightarrow E + I: \qquad k_{dI} \qquad (4)$$
$$K_i = k_{dI}/k_{aI}$$

where the bimolecular association rate constant $k_{aI}$, is assumed to be $1.0 \, \mu M^{-1}s^{-1}$.

### SARS-CoV-2 S protein cleavage inhibition assays
The SARS-CoV-2 S protein construct HexaFurin was prepared as a substrate for dasTMPRSS11D as previously demonstrated for dasTMPRSS2[15]. dasTMPRSS11D was pre-incubated with 100-0.01 $\mu$M nafamostat ($\log_{10}$ dilution series) or 1000-62.5 $\mu$M 6-amidino-2-naphthol ($\log_2$ dilution series) prior to transferring the enzyme-inhibitor mixtures to 5 $\mu$g S protein substrate. S protein cleavage assays were carried out over 20 min at room temperature before assays were quenched through addition of 4X Laemmli buffer. SDS-PAGE samples were thermally denatured (5 min at 95 °C) and approximately 4 $\mu$g S protein were loaded per well. After gel separation, protein bands were visualized by Coomassie blue staining.

### DSF for inhibitor-induced TMPRSS11D $\Delta T_m s$
TMPRSS11D protein $T_m s$ and ligand-induced $\Delta T_m s$ were measured using SYPRO Orange dye (Life Technologies, catalog no. S-6650) and SYPRO fluorescence at 470 and 510 nm excitation and emission, respectively, using the Light Cycler 480 II (Roche Applied Science). Samples were prepared in technical triplicate in 384-well plates (Axygen; catalog nos. PCR-384-C; UC500) at a final volume of 20 $\mu$L. Wells contained 2 $\mu$g TMPRSS11D protein, 10% ($v/v$) ligand (or DMSO control) and 5× SYPRO Orange. Thermal melt curves were generated across a 25–95 °C gradient at a heating rate of 2 °C/min. TMPRSS11D $T_m$ values +/− s.d. were calculated using the dRFU method with the DSFworld application[65]. Ligand-induced dasTMPRSS11D or dasTMPRSS11D S368A $\Delta T_m s$ relative to DMSO control were calculated for nafamostat or 6-amidino-2-naphthol and plotted using Graphpad Prism.

### Data exclusion and statistics
The two-sided Grubbs' test was used to determine and exclude single outliers present in sample data performed in triplicate or greater.

Single datapoint outliers were identified in Fig. 3d ($n = 3$ samples, 0 $\mu$M nafamostat control single replicate excluded); Supplementary Fig. 2 ($n = 8$ samples, 1000 $\mu$M Boc-QAR-AMC single replicate excluded, 0.488 $\mu$M Boc-QAR-AMC two replicates excluded). Within the supplied Source Data file, these exclusions are denoted with blue text and asterisks.

### Reporting summary
Further information on research design is available in the Nature Portfolio Reporting Summary linked to this article.

## Data availability
The coordinates and structures of dasTMPRSS11D complexed with a DDDDK peptide and eTMPRSS11D S368A complexed with a LSEQR peptide have been deposited to the PDB with accession numbers 8VIS and 9DPF, respectively. The engineered dasTMPRSS11D and eTMPRSS11D S368A protein expression constructs are available on Addgene (Plasmid nos. 220881 and 220882, respectively). Zymogen matriptase, active matriptase, hepsin, TMPRSS2, bovine enteropeptidase, human enteropeptidase, TMPRSS11E, TMPRSS13, and mouse TMPRSS11D SEA domain structures were accessed using PDB IDs 5LYO, 4JYT, 1Z8G, 7MEQ & 8V04, 1EKB, 7WR7, 2OQ5, 6KD5, and 2E7V, respectively. Source data are provided as a Source Data file. Source data are provided with this paper.

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

## Acknowledgements

This work was supported by a BC Leadership Chair in Functional Cancer Imaging to F.B., and a Killam Doctoral Fellowship and a Mitacs Elevate Postdoctoral Fellowship to B.J.F. We thank Evianne Rovers for her assistance in the AlphaFold model quality control check and the Predicted Aligned Error plot. The synthesis of peptidomimetic inhibitors was supported by a Canada Institute of Health Research (CIHR) grant (PJT-183813) to P.-L.B and R.L. and the production of recombinant Spike protein for biochemical assays was supported by an NIH grant 1U19AI171292-01 (READDI-AViDD Center) to C.H.A. This work is based upon research conducted using beamline CMCF-ID at the Canadian Light Source, a national research facility of the University of Saskatchewan, which is supported by the Canada Foundation for Innovation (CFI), the Natural Sciences and Engineering Research Council (NSERC), the National Research Council (NRC), the Canadian Institutes of Health Research (CIHR), the Government of Saskatchewan, and the University of Saskatchewan. The Structural Genomics Consortium is a registered charity (no: 1097737) that receives funds from AbbVie, Bayer AG, Boehringer Ingelheim, Genentech, Genome Canada through Ontario Genomics Institute [OGI-196], the EU and EFPIA through the Innovative Medicines Initiative 2 Joint Undertaking [EUbOPEN grant 875510], Janssen, Merck KGaA (aka EMD in Canada and US), Pfizer, Takeda and the Wellcome Trust [106169/ZZ14/Z].

## Author contributions

C.H.A., B.J.F., G.B.M., F.B., P.-L.B., R.L., and L.Z.P. provided project supervision; B.J.F conceived the project; B.J.F. and R.P.W. designed the experiments; Y.L. cloned TMPRSS2 and TMPRSS11D protein constructs for expression; A.S., R.P.W., O.I., Y-Y.L., and Z.H. produced TMPRSS2 and TMPRSS11D proteins in insect cells; B.J.F., O.I., and J.L. purified recombinant proteins; B.J.F. crystallized TMPRSS11D and B.J.F. and A.D. collected diffraction data; B.J.F, A.D., and T.M.G.K. solved the TMPRSS11D crystal structures; S.F. synthesized, purified and chemically characterized peptidomimetic compounds; B.J.F. performed bioinformatic and structural analyses; B.J.F., R.P.W. and J.L. performed fluorogenic peptidase activity and inhibitor potency assays and analyzed kinetics; B.J.F. and O.I. performed gel-based S protein digestion assays; B.J.F., O.I., and J.L. performed DSF assays; B.J.F. prepared figures; B.J.F., R.P.W., C.H.A., S.F., G.B.M., R.L., and A.M.E. wrote the manuscript.

## Competing interests

The authors declare no competing interests.
