## [Transparent Peer Review file · Nature Communications]

Structural basis of TMPRSS11D specificity and autocleavage activation

Corresponding Author: Professor Cheryl Arrowsmith

Version 0:

Reviewer comments:

Reviewer #1

(Remarks to the Author)

Fraser and colleagues study the structural basis of the activation of TMPRSS2 and TMPRSS11D serine proteases. They report that the proteases cleave their zymogen activation motifs. They solved a TMPRSS11D co-crystal structure in complex with a native TMPRSS11D zymogen activation motif. The protease inhibitor Nafamostat mesylate is cleaved by TMPRSS11D and converted to low activity derivatives. I am not a structuralist and I will not comment in detail the results. As a non-specialist, I found experiments well performed. The manuscript is clearly written, providing original insights into in vitro serine protease autocleavage activation.

1. It would have been interesting to ask whether the proteases are similarly activated at the cell surface and to examine the spike cleavage by TMPRSS11D and TMPRSS2 and its inhibition by small molecule inhibitors in a cellular context. This could be at least discussed. The absence of any in cellulo experiments should be presented as a limitation of the study.

Reviewer #2

(Remarks to the Author)

Fraser et al. demonstrate that human membrane bound serine proteases TMPRSS2 and TMPRSS11D can autoproteolyse their own zymogen activation motifs, and they determine the crystal structure of TMPRSS11D with the native and engineered zymogen activation motifs, revealing substrate interactions in the active site. Furthermore, they demonstrate that the TMPRSS2 inhibitor Nafamostat is turned over as a substrate by TMPRSS11D. This work points to different specificities of both proteases, which can be of use in further research of these pharmacologically interesting enzymes.

The work rests on an observation made and published previously by these authors, that expression of native TMPRSS2 in insect cells is negligible (presumably because it is toxic), while mutation of the zymogen activation cleavage site in TMPRSS2 to a less effective one allows production of the soluble ectodomain of the proenzyme capable of auto-activation at high concentration to produce active TMPRSS2. This concentration dependence indicates that the cleavage occurs in trans, i.e. is intermolecular. Expectedly, this principle works also for TMPRSS11D, which the authors show in the present manuscript, and is thus not novel.

Using this approach, the authors then determine crystal structures of TMPRSS11D in the zymogen-cleaved form and find that the disulfide-linked zymogen activation motif (its non-prime side) mediates crystal contacts between the adjacent TMPRSS11D molecule in the crystal lattice by binding into its active site. This allows visualization of the substrate interactions with TMPRSS11D active site at the non-prime side, which can be of use for the structure-assisted design of selective inhibitors of this enzyme.

This work presents very good crystallography of a relevant protein, but does not have obvious implications for the biology of TMPRSS11D or TMPRSS2. It also falls short of using of the gained structural information for the design of molecules that would exploit it and show selectivity between these two enzymes (and perhaps some others of the mentioned TMPRSS family members, such as those shown in Fig. 4 and 5). There are a number of pharmacophores that bind covalently and reversibly to serine proteases (such as aldehydes, boronates, phosphonates, acyl sulfonamides, thiazolyl ketones, ketoamides, ...) and can be decorated by short substrate-derived peptides, which could be used as a basis to demonstrate

the claimed differences in active site preferences.

In summary, this work is of excellent technical quality, but lacks conclusive novelty and is thus suited for a more specialised journal.

Version 1:

Reviewer comments:

Reviewer #1

(Remarks to the Author)

The authors have addressed my concerns.

Reviewer #2

(Remarks to the Author)

Fraser et al. provide a revised version of their manuscript in which they synthesised 6 ketobenzothiazoles and a minimal SAR on these. While they were able to design selective TMPRSS2 inhibitor and an inhibitor of both TMPRSS11D or TMPRSS2. Although a TMPRSS11D selective inhibitor has not been designed yet, this study now more satisfactorily exploits its structural insights into TMPRSS11D, and it provides a robust platform upon which future efforts into structure-assisted design of selective inhibitors of TMPRSS family members can be based. Discussion now eloquently illustrates the broader significance of this for the biology of TMPRSS activation and for drug discovery targeting this important family of enzymes. I have only a few really minor observations:

1. section "TMPRSS2 and TMPRSS11D are inhibited by peptides mimicking their activation motifs."

- typo in "derivitized" should read "derivatized"

2. section

- "These biophysical data confirm that nafamostat relies on the TMPRSS11D S368A residue to induce.." should be probably corrected to "These biophysical data confirm that nafamostat relies on the TMPRSS11D S368 residue to induce..."

3. In general, inhibitors whose IC50 approaches the concentration of the enzyme (or half of it), the IC50 values are getting less sensitive for comparison of their potency as tight-binding conditions are approached. This IC50 area may be the case of some of the inhibitors, provided that the enzyme concentration being reported accurately reflects the real concentration of active enzyme. How were concentrations of active TMPRSS-s determined?

Reviewer #1 (Remarks to the Author):

1. *It would have been interesting to ask whether the proteases are similarly activated at the cell surface and to examine the spike cleavage by TMPRSS11D and TMPRSS2 and its inhibition by small molecule inhibitors in a cellular context. This could be at least discussed. The absence of any in cellulo experiments should be presented as a limitation of the study.*

To contextualize the biochemical and biophysical data presented in this work characterizing the molecular basis of TMPRSS2 and TMPRSS11D autocleavage activation and respiratory virus entry protein cleavage, we have revised several portions of the results and discussion text and highlight previous, cell-based biological studies of these proteases that support our findings.

Reviewer 1: *“It would have been interesting to ask whether the proteases are similarly activated at the cell surface”*

Response: Specifically, after we present our findings that soluble, concentrated dasTMPRSS11D enzyme had produced a protein band migrating at ~27 kDa (lines 107-112), we reference the work of Kato et al. (2012) and Böttcher-Friebertshäuser et al. (2010). In their studies, they found that full-length TMPRSS11D protein overexpressed in HEK293 and MDCK cells produced a 27 kDa protein band (detected with antibodies specific for the recombinant TMPRSS11D protein C-terminus) that is shed into cell media. Our recombinant TMPRSS11D protein evidently undergoes autocleavage activation, and we further show that we can use an active TMPRSS11D molecule to fully activate immature TMPRSS11D zymogen proteins, confirming an *in-trans* activation mechanism.

Reviewer 1: *“It would have been interesting.... to examine the spike cleavage by TMPRSS11D and TMPRSS2 and its inhibition by small molecule inhibitors in a cellular context. This could be at least discussed.”*

Response: We provide additional references to cell-based studies of respiratory virus infection that used small molecule protease inhibitors to disable TMPRSS2 and TMPRSS11D. Specifically, after describing our recombinant TMPRSS11D spike cleavage results and blockade by nafamostat and 6-amidino-2-naphthol, we note that “ These data match previous findings that competitive trypsin-like serine protease inhibitors can block TMPRSS11D-driven H1N1 viral entry to MDCK cells (Bottcher et al. 2009; Bottcher-Friebertshäuser et al. 2010)”.

Reviewer 1: *The absence of any in cellulo experiments should be presented as a limitation of the study.*

Response: In the discussion text, we note that lack of cellular data is a limitation of our study. This is addressed by the statement “Although evidence suggests that these proteases undergo autocleavage activation in cells^{14,28,45}, we have not yet evaluated their potential to facilitate cross-activation in cell-based systems in this study.”

Reviewer 2: *This work presents very good crystallography of a relevant protein, but does not have obvious implications for the biology of TMPRSS11D or TMPRSS2. It also falls short of using of the gained structural information for the design of molecules that would exploit it and show selectivity between these two enzymes (and perhaps some others of the mentioned TMPRSS family members,*

such as those shown in Fig. 4 and 5). There are a number of pharmacophores that bind covalently and reversibly to serine proteases (such as aldehydes, boronates, phosphonates, acyl sulfonamides, thiazolyl ketones, ketoamides, ...) and can be decorated by short substrate-derived peptides, which could be used as a basis to demonstrate the claimed differences in active site preferences.

Response: We developed five ketobenzothiazole-containing peptidomimetic inhibitor compounds targeting the substrate binding cleft of TMPRSS11D and TMPRSS2: **1** (Ac-Glu-Gln-Arg-kbt), **2** (Ac-Gln-Ser-Arg-kbt), **3** (Ac-Glu-Glu-Arg-kbt), **4** (Ac-Glu-D-Glu-Arg-kbt), and **5** (Ac-Glu-Orn-Arg-kbt). Peptidomimetics **1** and **2** were based upon the zymogen activation motif of TMPRSS11D and TMPRSS2, respectively, and both were potent, nanomolar inhibitors of TMPRSS11D and TMPRSS2. To improve TMPRSS11D binding affinity, we installed modifications to **1**, producing peptidomimetics **3-5**. This approach was chosen because peptidomimetic **1** closely resembled the peptide ligand in our TMPRSS11D crystal structure (LSEQR-CO₂⁻) and allowed us to have a structure-guided interpretation of the biochemical inhibition data. We outline the results of these Structure-Activity-Relationship studies and impacts on protease selectivity in our revised results and discussion sections:

i) RESULTS SECTIONS

TMPRSS2 and TMPRSS11D are inhibited by peptides mimicking their activation motifs.

We developed *in vitro* assays to evaluate TMPRSS11D inhibitors, laying the groundwork for structure-activity relationship studies and drug discovery. For inhibitor screening, we selected the Boc-QAR-AMC substrate over Boc-QRR-AMC which had TMPRSS11D K_M values of 8.3 and 181 μM, respectively (Extended Data Fig. 2a-b). Since TMPRSS11D and TMPRSS2 efficiently cleave their own zymogen motifs, the C-terminus of peptides containing parts of the TMPRSS11D and TMPRSS2 zymogen motifs were derivitized with ketobenzothiazole warheads, creating Ac-Glu-Gln-Arg-kbt (**1**) and Ac-Gln-Ser-Arg-kbt (**2**) inhibitory peptides, respectively (Fig. 2e). Both **1** and **2** were potent TMPRSS11D inhibitors with respective half-maximal inhibitory concentrations (IC₅₀s) of 29 [24, 34] nM and 33 [26,41] nM (95% confidence interval reported in brackets; Fig. 2f). Nafamostat, camostat and 6-amidino-2-naphthol, which are covalent and competitive inhibitors of trypsin-like serine proteases, had much weaker TMPRSS11D IC₅₀ values of 55 [48,65] nM, 107 [93, 122] nM and 37 [30, 46] μM, respectively. These data suggested that TMPRSS11D was more potently inhibited by kbt-containing peptides than traditional small molecule trypsin-like serine protease inhibitors. In contrast TMPRSS2 was inhibited by **1**, **2**, nafamostat and camostat with similarly potent IC₅₀ values of 4.6 [4.3, 5.1] nM, 2.9 [2.7, 3.1] nM, 1.6 [1.2, 1.8] nM, and 5.5 [5.2, 5.9] nM respectively, but was weakly inhibited by 6-amidino-2-naphthol with an IC₅₀ value of 5.5 [5.0, 6.1] μM (Fig. 2f). Our *in vitro* assays identified ketobenzothiazole-containing peptides as potent TMPRSS11D inhibitors, outperforming traditional small-molecule inhibitors, while TMPRSS2 inhibition remained consistent across peptides and nafamostat.

The TMPRSS11D and TMPRSS2 S2 are distinct and targetable with peptide ligands

The S2 of TMPRSS11D contains a unique combination of R230, Y264, and H269 which is distinct from every other TTSP whereas the TMPRSS11D S3 closely resembled the TMPRSS11E S3 (Fig. 5b,e-f). We therefore focused our attention on identifying favorable ligand interactions with this site. TMPRSS2 also contains a positively charged amino acid within its S2, K342 as well as a negatively charged E299 at the equivalent position of TMPRSS11D's R230. Notably, R230 did not make direct interactions with a P2

Gln or P2 Asp in our TMPRSS11D structures (Fig. 5e), suggesting that improvements could be made by modifying the P2 amino acid. In our biochemical assays, peptidomimetics **1** and **2** containing P2 Gln and Ser, respectively, were tolerated by the S2 of TMPRSS11D and TMPRSS2. Similarly, substrate peptide Boc-QAR-AMC containing a P2 Ala was efficiently cleaved by TMPRSS11D and TMPRSS2, whereas Boc-QRR-AMC containing a P2 Arg was a poor substrate for both proteases.

To probe if TMPRSS11D's R230 residue could be pharmacologically targeted, we prepared three analogs of peptidomimetic **1** with P2 substitutions, Ac-Glu-Glu-Arg-kbt (Ac-EER-kbt; **3**), Ac-Glu-D-Glu-Arg-kbt (Ac-EeR-kbt; **4**), and an ornithine (Orn)-containing peptidomimetic Ac-Glu-Orn-Arg-kbt (**5**). Peptidomimetic **3** was the most potent TMPRSS11D inhibitor identified with an IC₅₀ of 5.3 [4.2, 6.8] nM. Peptidomimetic **4** with a P2 D-Glu had moderate TMPRSS11D binding affinity with an IC₅₀ of 117 [94, 146] nM. Peptidomimetic **5** containing a positively charged P2 Orn lost substantial TMPRSS11D binding affinity relative to all other peptidomimetics, with a TMPRSS11D IC₅₀ value of 2.3 [1.9, 2.8] μM. For TMPRSS2, peptidomimetics **3** and **5** retained potent binding affinity with TMPRSS2 IC₅₀ values of 2.2 [2.1, 2.6] nM and 9.6 [8.7, 10.5] nM, respectively, whereas peptidomimetic **5** had moderate binding affinity with an IC₅₀ value of 52 [47, 57] nM. These results highlight the importance of P2 residue selection in modulating inhibitor potency and selectivity, with peptidomimetic **3** emerging as a highly potent dual TMPRSS11D and TMPRSS2 inhibitor, likely due to favorable electrostatic interactions. In contrast, peptidomimetic **5** exhibited selectivity for TMPRSS2 over TMPRSS11D, suggesting that electrostatic repulsion at TMPRSS11D's R230 contributes to its reduced binding affinity.

ii)DISCUSSION SECTION

Using the eTMPRSS11D S368A crystal structure complexed with the LSEQR-CO₂⁻ peptide ligand, we mapped the S1'-S3 subsites of TMPRSS11D, providing an opportunity to rationally target the TMPRSS11D binding cleft with peptidomimetic inhibitors for the first time, focusing on exploring S2 interactions in the current study. The S2 of TMPRSS11D contains a unique combination of R230 and H269, distinct from every other TTSP. The S2 of TMPRSS2, by contrast, contains E299 and K342 at the equivalent positions. Both proteases accommodated a P2 Gln residue, as evidenced by potent inhibition by peptidomimetic **1**, but showed weaker affinity than peptidomimetic **2**, which contained a P2 Ser residue. Analogs **3-5** explored the effects of incorporating P2 Glu, D-Glu, and Orn residues on TMPRSS11D and TMPRSS2 binding affinity. A P2 Glu yielded the most potent reported TMPRSS11D inhibitor, **3**, with an IC₅₀ of 5.3 [4.2, 6.8] nM. Peptidomimetic **3** was also the most potent TMPRSS2 inhibitor in this study, with an IC₅₀ of 2.2 [2.1, 2.6] nM. In contrast, introducing a positively charged Orn (peptidomimetic **5**) selectively inhibited TMPRSS2 with an IC₅₀ of 9.6 [8.7, 10.5] nM but substantially reduced TMPRSS11D binding affinity (IC₅₀ = 2.3 [1.9, 2.8] μM). Peptidomimetic **4**, containing a P2 D-Glu, retained moderate binding affinity for both TMPRSS11D (IC₅₀ = 117 [94, 146] nM) and TMPRSS2 (IC₅₀ = 52 [47, 57] nM). This moderate binding affinity, despite the introduction of a D-amino acid, could be explained by continued salt bridge and/or H-bond interactions with the S2 residues of TMPRSS11D and TMPRSS2 but induced a new conformation of the P3 ligand residue. We therefore identified a potent dual TMPRSS11D and TMPRSS2 inhibitor (**3**), a TMPRSS2-selective inhibitor (**5**), and a unique D-amino acid-containing peptidomimetic with moderate affinity for both proteases (**4**). Unlike TMPRSS11D, TMPRSS2 has been widely targeted with peptidomimetic inhibitors, but to our knowledge, P2 Glu, D-Glu, or Orn have not previously been explored as scaffolds for TMPRSS2 inhibition. Further structure-guided optimization of peptidomimetics **3-5**, particularly through substitution of the P3 ligand residue, may enhance potency and selectivity towards TMPRSS11D and/or TMPRSS2 by enhancing complementarity with their substrate binding clefts.

Reviewer #2 (Remarks to the Author):

1. Fraser et al. provide a revised version of their manuscript in which they synthesised 6 ketobenzothiazoles and a minimal SAR on these. While they were able to design selective TMPRSS2 inhibitor and an inhibitor of both TMPRSS11D or TMPRSS2. Although a TMPRSS11D selective inhibitor has not been designed yet, this study now more satisfactorily exploits its structural insights into TMPRSS11D, and it provides a robust platform upon which future efforts into structure-assisted design of selective inhibitors of TMPRSS family members can be based. Discussion now eloquently illustrates the broader significance of this for the biology of TMPRSS activation and for drug discovery targeting this important family of enzymes. I have only a few really minor observations:

1. section "TMPRSS2 and TMPRSS11D are inhibited by peptides mimicking their activation motifs." - typo in "derivitized" should read "derivatized"

Response: Typo has been corrected

2. section- "These biophysical data confirm that nafamostat relies on the TMPRSS11D S368A residue to induce.." should be probably corrected to "These biophysical data confirm that nafamostat relies on the TMPRSS11D S368 residue to induce..."

Response: Typo has been corrected

3. In general, inhibitors whose IC50 approaches the concentration of the enzyme (or half of it), the IC50 values are getting less sensitive for comparison of their potency as tight-binding conditions are approached.

Response: IC50 values for some of the inhibitors approached half the enzyme concentrations for dasTMPRSS2 (nafamostat, PM-1, PM-2, PM-3) and dasTMPRSS11D (PM-2, PM-3). However, the IC50 values for these compounds (and differences between IC50s for different compounds) were reproducible across $n > 4$ independent biological replicates and are informative for SAR. Importantly, no compounds in this study had the same IC50 values and indicate that the tight-binding conditions for peptidomimetic compounds were not reached here (but most likely were for TMPRSS2-nafamostat).

This IC50 area may be the case of some of the inhibitors, provided that the enzyme concentration being reported accurately reflects the real concentration of active enzyme. How were concentrations of active TMPRSS-s determined?

Response: The Methods section now includes additional experimental details outlining how the concentrations of TMPRSS2 and TMPRSS11D protein were determined:

“Protein concentration was initially estimated using absorbance at 280 nm (A_{280}) with molar extinction coefficients of $52,940 \text{ M}^{-1}\text{cm}^{-1}$ and $99,350 \text{ M}^{-1}\text{cm}^{-1}$ for the catalytic domain of dasTMPRSS11D and the ectodomain of dasTMPRSS2, respectively. For dasTMPRSS2, the enzyme concentration in the assay was

determined as 2-fold the nafamostat IC_{50} value due to its exceptional inhibitory potency and utility as a TMPRSS2 burst titrant. For dasTMPRSS11D, the enzyme concentration in the assay was determined through reaction progress curve fitting for various dasTMPRSS11D enzyme dilutions incubated with saturating concentrations of Boc-QAR-AMC substrate ($100 \mu\text{M}$; Boc-QAR-AMC $K_m = 8 \mu\text{M}$), with curve-fitted data available in Supplementary Figure 3a.”

Supplementary Figure 3. TMPRSS11D enzyme kinetic models to determine kinetic inhibition parameters for 6-amidino-2-naphthol and nafamostat. a, TMPRSS11D Boc-QAR-AMC substrate conversion rate, k_{sub} . Reaction progress curves were generated with the indicated concentrations of dasTMPRSS11D enzyme and $100 \mu\text{M}$ Boc-QAR-AMC substrate and monitored for AMC product formation. Progress curves were fitted across all enzyme

concentrations to determine the k_{sub} parameter (Model 1; Methods). Datapoints and curve fits are shown, with data residuals (curve fit value – experimental data) plotted below. **b**, TMPRSS11D inactivation potency parameter, k_{inact}/K_i , with nafamostat. Progress curves were generated using 3 nM dasTMPRSS11D, 100 μM Boc-QAR-AMC substrate, and the indicated concentrations of nafamostat. Inhibitor and substrate were simultaneously transferred to wells containing dasTMPRSS11D and fluorescence was monitored immediately. Data curve-fitted according to Model 2 (Methods). **d**, dasTMPRSS11D nafamostat inhibition half-life parameter, $t_{1/2}$. Data was generated using 3 nM dasTMPRSS11D and 100 μM Boc-QAR-AMC substrate. Nafamostat (at the indicated concentrations) was incubated with dasTMPRSS11D for 3 minutes prior to substrate transfer. Data was curve-fitted to determine the hydrolysis rate of the dasTMPRSS11D:nafamostat acyl-enzyme complex (Model 3). **d**, dasTMPRSS11D inhibition constant (K_i) for 6-amidino-2-naphthol. Progress curves were generated using 3 nM dasTMPRSS11D, 100 μM Boc-QAR-AMC substrate in the same assay protocol as (**b**). Data was curve-fitted according to Model 4 (Methods).